# SCube: Instant Large-Scale Scene Reconstruction using VoxSplats

**Xuanchi Ren**[1,2,3*]**, Yifan Lu**[1,4*]**, Hanxue Liang**[1,5]**, Zhangjie Wu**[1,6]**,
Huan Ling**[1,2,3]**, Mike Chen**[1]**, Sanja Fidler**[1,2,3]**, Francis Williams**[1]**, Jiahui Huang**[1]

[1]NVIDIA, [2]University of Toronto, [3]Vector Institute, [4]Shanghai Jiao Tong University
[5]University of Cambridge, [6]National University of Singapore
https://research.nvidia.com/labs/toronto-ai/scube/

## Abstract

We present SCube, a novel method for reconstructing large-scale 3D scenes (geometry, appearance, and semantics) from a sparse set of posed images. Our method encodes reconstructed scenes using a novel representation VoxSplat, which is a set of 3D Gaussians supported on a high-resolution sparse-voxel scaffold. To reconstruct a VoxSplat from images, we employ a hierarchical voxel latent diffusion model conditioned on the input images followed by a feedforward appearance prediction model. The diffusion model generates high-resolution grids progressively in a coarse-to-fine manner, and the appearance network predicts a set of Gaussians within each voxel. From as few as *3 non-overlapping input images*, SCube can generate millions of Gaussians with a $1024^3$ voxel grid spanning *hundreds of meters* in *20 seconds*. Past works tackling scene reconstruction from images either rely on per-scene optimization and fail to reconstruct the scene away from input views (thus requiring dense view coverage as input) or leverage geometric priors based on low-resolution models, which produce blurry results. In contrast, SCube leverages high-resolution sparse networks and produces sharp outputs from few views. We show the superiority of SCube compared to prior art using the Waymo self-driving dataset on 3D reconstruction and demonstrate its applications, such as LiDAR simulation and text-to-scene generation.

## 1 Introduction

Recovering 3D geometry and appearance from images is a fundamental problem in computer vision and graphics which has been studied for decades. This task lies at the core of many practical applications spanning robotics, autonomous driving, and augmented reality; just to name a few. Early algorithms tackling this problem use stereo matching and structure from motion (SfM) to recover 3D signals from image data (*e.g.*[44]). More recently, a line of work starting from Neural Radiance Fields [32] (NeRFs) has augmented traditional SfM pipelines by fitting a volumetric field to a set of images, which can be rendered from novel views. NeRFs augment traditional reconstruction pipelines by encoding dense geometry, and view-dependent lighting effects. While radiance-field methods present a drastic step forward in our ability to recover 3D information from images, they require a time-consuming per-scene optimization scheme. Furthermore, since each scene is recovered in isolation, radiance fields do not make use of data priors, and cannot extrapolate reconstructions away from the input views. Thus, radiance-field methods require dense view coverage in order to produce high-quality 3D reconstructions.

---

[*]Equal contribution.

38th Conference on Neural Information Processing Systems (NeurIPS 2024).

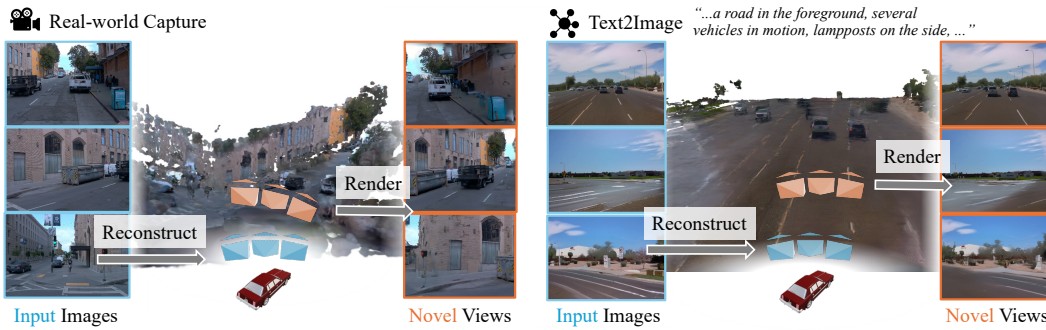

Figure 1: **SCube.** Given sparse input images with little or no overlap, our model reconstructs a high-resolution and large-scale scene in 3D represented with VoxSplats, ready to be used for novel view synthesis or LiDAR simulation.

Another recent line of work applies deep learning to predict 3D from images. These methods either meta-learn an initialization to the radiance-field optimization problem [7, 30, 49], or directly predict 3D from images using a feed-forward network [17, 57, 73]. While learning-based approaches can produce reconstructions from sparse views, they have only been used successfully for the case of single objects at low resolutions. Furthermore, these methods often suffer from 3D inconsistencies (*e.g.* the multi-layer surface or the Janus problem). In order to solve the general 3D reconstruction from images problem, we need methods that can *(1)* generalize reconstruction to general scenes over the pure object case, *(2)* produce accurate and high-quality reconstructions in the presence of dense views, leveraging data priors to produce plausible reconstructions in the sparse-view regime, and *(3)* run quickly and efficiently (in terms of runtime and memory) on large-scale and high-resolution inputs. These demands are difficult to satisfy in practice since high-quality ground-truth 3D data is not widely available for scenes, 3D representations for deep learning that scale to large and diverse inputs are under-explored in the literature, and corresponding scalable and easy-to-train model designs need to be developed alongside any new 3D representation.

Nevertheless, we remark that some of these issues have been resolved in isolation: Gaussian Splatting [23] enables fast, differentiable rendering and high reconstruction quality (but is not being used with data priors), and sparse voxel hierarchies [40] have been successfully used to build generative models of large-scale 3D scenes with attributes such as semantics and colors, and have been trained on partial data such as LiDAR scans from autonomous vehicle captures.

In light of the above observations, we introduce SCube, a feed-forward method for large 3D scene reconstruction from images. Our method encodes 3D scenes as a hybrid of Gaussian splats (which enable fast rendering), supported on a sparse-voxel-hierarchy (which enables efficient generative modeling of large 3D scenes with semantics). We call this hybrid representation VoxSplats and predict a VoxSplat from images using a feed-forward process consisting of two steps: *(1)* A generative geometry network that predicts a sparse voxel hierarchy conditioned on input images, and *(2)* an appearance network that predicts the Gaussian attributes within the voxels as well as a skybox texture to represent the background. The networks are implemented using highly efficient sparse convolution [14, 40] designed for 3D data which enables us to reconstruct a full scene from images in under 20 seconds. We evaluate our performance on the Waymo Open Dataset [53] on the challenging task of reconstructing a scene from sparse images with low overlap. We show that SCube significantly outperforms existing methods on this task. Furthermore, we demonstrate that SCube enables downstream applications such as LiDAR simulation and text-to-scene generation.

## 2  Related Work

**3D Scene Representation.**  Scenes in the wild are often large in scale and contain complicated internal structures which cause representations such as tri-planes [12], dense voxel grids [36], or meshes [19, 46] to fail due to capacity or memory limitations. Optimization-based reconstruction methods [15, 32] use high-resolution hash grids [1, 33], but these are non-trivial to infer using a neural

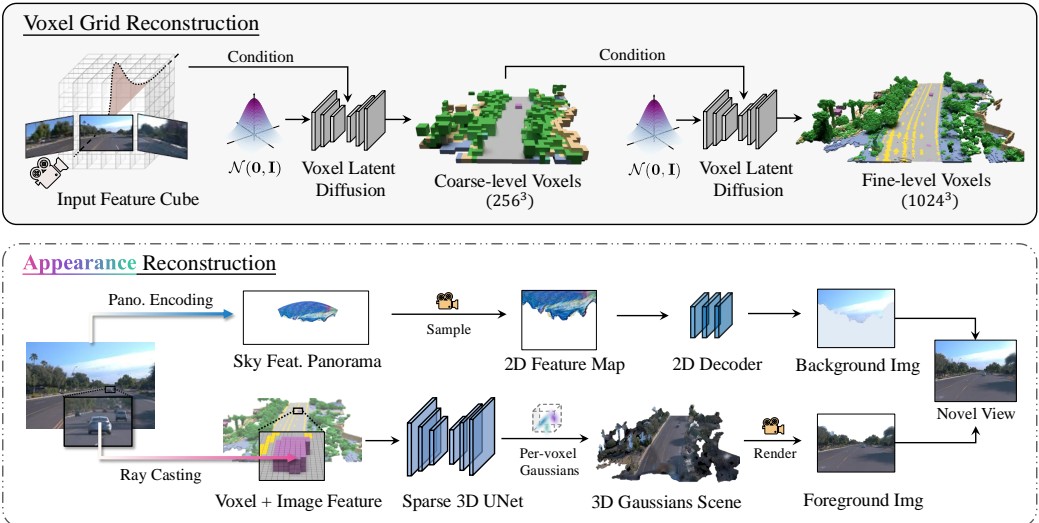

Figure 2: **Framework.** SCube consists of two stages: **(1)** We reconstruct a sparse voxel grid with semantic logit conditioned on the input images using a conditional latent diffusion model based on XCube [40]. **(2)** We predict the appearance of the scene represented as VoxSplats and a sky panorama using a feedforward network. Our method allows us to synthesize novel views in a fast and accurate manner, along with many other applications.

network [30]. In contrast, sparse voxel grids are effective for learning scene-reconstruction [40, 75] thanks to efficient sparse neural operators [8, 54]. Recently, Gaussian splatting [23] has enabled real-time neural rendering and has been applied to overfitting large scenes [66, 76]. [31, 39] use a hybrid of the above two representations, but the voxel grid or octree is only used to regularize the Gaussian positions without any data priors learned. This is in contrast to our VoxSplat that allows reconstruction in a direct inference pass thanks to the efficiency of sparse grids and the high representation power of Gaussian splats. We support operating only on sparse-view images, significantly lifting the input requirements by learning from large datasets.

**Sparse-view 3D Reconstruction.** Sparse-view images often contain insufficient correspondences required by traditional reconstruction methods [44]. One line of work uses learned image-space priors such as depth [9], normal maps [70], and appearance from GANs [43] or diffusion models [63] to augment an optimization process such as NeRF. To speed up inference, another line of work uses a feed-forward model to predict renderable features [4, 6, 22, 29, 57, 73]. Alternatively, some papers perform learning directly in 3D space, which yields better consistency and less distortion [5, 13, 17, 68]. Our setting is similar to [13] where input images come from the same rig, but ours is more challenging since we do not use temporally-sequenced inputs with high overlap. We remark that semantic scene completion works [21, 26, 51, 61] also reconstruct voxels but at much lower resolutions and without appearance.

**Generative Models for 3D.** 3D reconstruction can also be formulated as a conditional generative problem (i.e. modeling the distribution of scenes given partial observations). Text and single-image to-3D generation has been explored for objects [17, 28, 38, 47, 55, 56, 60, 69]. Extending this task to large-scenes is comparatively unexplored, and object-based methods often fail due to scaling limitations or assumptions on the data. [48, 72] recursively apply an image generative model to inpaint missing regions in 3D, but produces blurry reconstructions at a limited scale. XCube [40] is among the first to directly learn high-resolution 3D scene priors. Here, we extend this model with multiview image conditioning and enable it to predict appearance on top of geometry.

## 3 Method

Our method reconstructs a high-resolution 3D scene from $N$ sparse images $\mathcal{I} = \{\mathbf{I}^i\}_{i=1}^N$ in two stages: *(1)* We reconstruct the scene geometry represented as a sparse voxel grid $\mathcal{G}$ with semantic

features (§ 3.1). *(2)* We predict the appearance $\mathcal{A}$ of the scene that allows for high-quality novel view synthesis (§ 3.2) using VoxSplats and a sky panorama. We can express our pipeline as taking samples from the distribution $p(\mathcal{G}, \mathcal{A}|\mathcal{I}) = p(\mathcal{A}|\mathcal{G}, \mathcal{I})p(\mathcal{G}|\mathcal{I})$. In order to improve the final view quality of the output, we apply an optional post-processing step discussed in § 3.3.

## 3.1 Voxel Grid Reconstruction

**Background: 3D Generation with XCube.** XCube [40] is a 3D generative model that produces high-quality samples for both objects and large outdoor scenes. XCube uses a hierarchical latent diffusion model to generate *sparse voxel hierarchies*, i.e., a hierarchy of sparse voxel grids where each fine voxel is contained within a coarse voxel. XCube learns a distribution over latent $\mathbf{X}$ encoded by a sparse structure Variational Autoencoder (VAE). Both the VAE and the diffusion model are instantiated with sparse convolutional neural networks [14], and can generate geometry at up to $1024^3$ resolution. We use XCube as the backbone for our geometry reconstruction module. We remark that while the original paper only focused on unconditional or text-conditioned generation, we introduce a novel image-based conditioning $\mathbf{C}$.

**Image Conditioned Geometry Generation.** To condition XCube on posed input images, we lift DINO-v2 [34] features computed on the input images to 3D space as follows. First, we use the pre-trained DINOv2 model to extract robust visual features for input images, and process the DINO feature using several trainable 2D conv layers to reduce the feature channel to $C + D$. We then split the channel $C + D$ into two parts for each pixel $j$ and input image index $i$: one part is a regular $C$-dimensional feature $\mathbf{F}^i_j$ and the other will be a $D$-dimensional Softmax-normalized vector $\theta^i_j \in \mathbb{R}^D$. Here $\theta^i_j$ can be viewed as a distribution over the depth of the corresponding pixel, and we follow a strategy similar to LSS [37] to unproject the images into a dense 3D voxel grid $\Omega$ where $v$ denotes the index of a voxel and $d \in [1, D]$ indexes the depth buckets:

$$\mathbf{F}^i_{jd} = \theta^i_{jd} \cdot \mathbf{F}^i_j, \quad \mathbf{C}_v = \sum_{(i,j,d)} \mathbf{F}^i_{jd} \in \mathbb{R}^C. \tag{1}$$

Note that we quantize the depth into $D$ bins dividing the range from a predefined $z_{\text{near}}$ to $z_{\text{far}}$. Unlike image-conditioning techniques used in object-level or indoor-level datasets where the camera frusta have significant overlap, our large-scale outdoor setting only takes sparse low-overlapping views captured from an ego-centric camera. Hence previous methods [28, 50, 52] that broadcast the same features to all the voxels along the rays corresponding to the pixel are not suitable here to precisely locate the geometries such as vehicles. The use of the weight $\theta$ allows us to handle occlusions effectively and produce a more accurate conditioning signal. After building $\mathbf{C}$, we directly concatenate it with the latent $\mathbf{X}$ and feed it into the XCube diffusion network as conditioning.

**Training and Inference.** Our training pipeline is similar to [40], where we first train a VAE to learn a latent space over sparse voxel hierarchies. We add semantic logit prediction as in [40] to the grid and empirically find that it helps the model to learn better geometry. Then we train the diffusion model conditioned on $\mathbf{C}$ using the following loss:

$$\mathcal{L} = \mathcal{L}_{\text{Diffusion}} + \lambda \mathcal{L}_{\text{Depth}}, \quad \mathcal{L}_{\text{Depth}} = \mathbb{E}_{\mathbf{X},i,j}\text{Focal}(\theta^i_j, [\theta^i_j]_{\text{gt}}), \tag{2}$$

where $\mathcal{L}_{\text{Diffusion}}$ is the loss for diffusion model training (see Appendix A for details). Focal$(\cdot)$ is the multi-class focal loss [27]. This additional depth loss is an explicit supervision to properly weigh the image features and encourage correct placement into the corresponding voxels. Due to the generative nature of XCube, we could learn the data prior to generate complete geometry even if some of the ground-truth 3D data is incomplete.

## 3.2 Appearance Reconstruction

**The VoxSplat Representation.** In the second stage, we fix the voxel grid $\mathcal{G}$ generated from the geometry stage and predict a set of Gaussian splats in each voxel to model the scene appearance. Gaussian splatting [23] is a powerful 3D representation technique that models a scene's appearance volumetrically as sum of Gaussians:

$$G(\boldsymbol{x}) = \text{RGB} \cdot \alpha \cdot e^{-\frac{1}{2}(\boldsymbol{x}-\boldsymbol{\mu})^\top \boldsymbol{\Sigma}^{-1}(\boldsymbol{x}-\boldsymbol{\mu})}, \tag{3}$$

where $\alpha \in [0, 1]$ is the opacity, $\boldsymbol{\mu} \in \mathbb{R}^3$ is the center of each Gaussian, and $\boldsymbol{\Sigma} = \mathbf{R}\mathbf{S}\mathbf{S}^\top\mathbf{R}^\top \in \mathbb{R}^{3\times3}$ is its covariance. The covariance matrix is factorized into a rotation matrix $\mathbf{R}$ parameterized by a quaternion $\mathbf{q}$ and a scale diagonal matrix $\mathbf{S} = \text{diag}(\boldsymbol{s})$. Each Gaussian additionally stores a color value RGB. Note that the original paper uses a set of SH coefficients for view-dependent colors, but we only use the $0^\text{th}$-order SH in our model (i.e., without view-dependency) which we found to be sufficient for sparse-view reconstruction.

While the original Gaussian Splatting paper and its follow-ups [23, 67, 71] propose many heuristics to optimize the positions of Gaussians for a given scene, we instead choose to predict $M$ Gaussians per-voxel using a feed-forward model. We limit the positions of the Gaussians within a neighborhood of their supporting voxels, thus preserving the geometric structure of the supporting grid. By grounding the splats on a voxel *scaffold*, our reconstructions achieve better geometric quality without resorting to heuristics. Fittingly, we dub our voxel-supported Gaussian splats *VoxSplats*.

The output of our network is $\{[\bar{\boldsymbol{\mu}}_v, \bar{\alpha}_v, \bar{\boldsymbol{s}}_v, \bar{\mathbf{q}}_v, \text{RGB}_v] \in \mathbb{R}^{14}\}_M$ for each voxel $v$. To compute the per-Gaussian parameters used for rendering we apply the following activations:

$$\boldsymbol{\mu}_v = r \cdot \tanh \bar{\boldsymbol{\mu}}_v + \text{Center}_v, \quad \alpha_v = \text{sigmoid}(\bar{\alpha}_v), \quad \boldsymbol{s}_v = \exp \bar{\boldsymbol{s}}_v, \quad \mathbf{R}_v = \text{quat2rot}(\bar{\mathbf{q}}_v), \quad (4)$$

where $\text{Center}_v$ is the centroid of the voxel $v$, and $r$ is a hyperparameter that controls the range of a Gaussian within its supporting voxel. Here, we set $r$ to three times the voxel size. We can efficiently render the Gaussians predicted by our model using rasterization [23] or raytracing [11].

**Sky Panorama for Background.** To capture appearance away from the predicted geometry, our model builds a sky feature panorama $\mathbf{L} \in \mathbb{R}^{H_p \times W_p \times C_p}$ from all input images, which can be considered as an expanded unit sphere with an inverse equirectangular projection. For each pixel in the panorama $\mathbf{L}$, we get its cartesian coordinate $\mathbf{P} = (x, y, z)$ on the unit sphere and project $\mathbf{P}$ to the image plane to retrieve the image feature; since only the view direction decides the sky color, we zero the translation part of the camera pose in the projection step. We also apply a sky mask to ensure the panorama only focuses on the sky region.

To render a novel viewpoint with its extrinsics and intrinsics, we recover the background appearance by sampling the sky panorama and decoding it into RGB values. For each camera ray from the novel view, we calculate its pixel coordinate on the 2D sky panorama $\mathbf{L}$ with equirectangular projection and get the sky features via trilinear interpolation, resulting in a 2D feature map for the novel view. We finally decode the 2D feature map with a CNN network to get the background image $\mathbf{I}_{\text{bg}}$, which will be alpha-composited with the foreground image from Gaussian rasterization:

$$\mathbf{I}_{\text{pred}}(u, v) = \mathbf{I}_{\text{GS}}(u, v) + (1 - \mathbf{T}(u, v)) \cdot \mathbf{I}_{\text{bg}}(u, v) \tag{5}$$

where $\mathbf{I}_{\text{GS}}(u, v)$ is the rendered image of Gaussians, $(u, v)$ indicates the pixel coordinate, and $\mathbf{T}(u, v)$ is the accumulated transmittance map of the Gaussians (see [23] for details).

**Architecture Details.** We predict the $(M \times 14)$-dimensional vector $\{[\bar{\boldsymbol{\mu}}_v, \bar{\alpha}_v, \bar{\boldsymbol{s}}_v, \bar{\mathbf{q}}_v, \text{RGB}_v]\}_M$ for each voxel via a 3D sparse convolutional U-Net which takes as input the sparse voxel grid $\Omega$ outputted by the geometry stage, where each voxel contains a feature sampled from the input images as follows: We process each input image $\mathbf{I}^i$ using a CNN to get the image feature, and then cast a ray from each image pixel into $\Omega$, accumulating the feature in the first voxel intersected by that ray. Voxels that are not intersected by any rays receive a zero feature vector.

For the sky panorama model, we use the same image feature as above. In the training stage, we set smaller $H_p$ and $W_p$ for faster training and lower memory usage; in the inference stage, we increase $H_p$ and $W_p$ to get a sharper and more detailed sky appearance.

Given a set of training images $\{\mathbf{I}_{\text{gt}}^i\}_i$ and sky masks $\{\mathbf{M}^i\}_i$ distinct from the inputs, we supervise the appearance model using the loss:

$$\mathcal{L} = \lambda_1 \mathcal{L}_1(\mathbf{I}_{\text{pred}}^i, \mathbf{I}_{\text{gt}}^i) + \lambda_2 \mathcal{L}_1(\mathbf{T}^i, \mathbf{M}^i) + \lambda_{\text{SSIM}}\mathcal{L}_{\text{SSIM}}(\mathbf{I}_{\text{pred}}^i, \mathbf{I}_{\text{gt}}^i) + \lambda_{\text{LPIPS}}\mathcal{L}_{\text{LPIPS}}(\mathbf{I}_{\text{pred}}^i, \mathbf{I}_{\text{gt}}^i), \tag{6}$$

where the training views $\mathbf{I}_{\text{gt}}^i$ are sampled from nearby 10 views of the input images; the predicted views $\mathbf{I}_{\text{pred}}^i$ and transmittance masks $\mathbf{T}^i$ are rendered using Eq (5); and $\mathcal{L}_{\text{LPIPS}}/\mathcal{L}_{\text{SSIM}}$ are perceptual and structural metrics defined in [74] and [59].

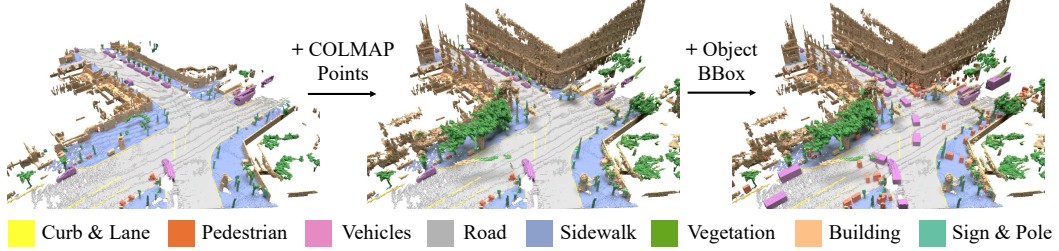

| Curb & Lane | Pedestrian | Vehicles | Road | Sidewalk | Vegetation | Building | Sign & Pole |

Figure 3: **Data Processing Pipeline.** We add COLMAP [44] dense reconstruction points to the accumulated LiDAR points and compensate for dynamic objects using their bounding boxes. This provides us with a more complete geometry for training.

### 3.3 Postprocessing and Applications

**Optional GAN Postprocessing.** The novel views directly rendered from our appearance model sometimes suffer from voxelization artifacts or noise. We resolve this with an optional lightweight conditional Generative Adversarial Network (GAN) that takes the rendered images as input and outputs a refined version. The discriminator of this GAN takes $256 \times 256$ image patches sampled from the input sparse view images, as well as the generated images conditioned on the rendered images. Drawing inspiration from [41, 43, 45], we fit the GAN independently for each scene at inference time, which takes ∼20min to train. Due to the excessive time cost, we apply this step optionally only when higher-quality images are needed (which we call **SCube+**). Fig. 8 shows the results with and without this step, and we further present a **general postprocessing without per-scene optimization** in Appendix C.

**Application: Consistent LiDAR Simulation.** LiDAR simulation [77] aims at reproducing the point cloud output given novel locations of the sensor and is an important application for training and verification of autonomous driving systems. The generated LiDAR point clouds should accurately reflect the underlying 3D geometry and a sequence of LiDAR scans should be temporally consistent. Our method enables converting sparse-view images directly into LiDAR point clouds, i.e., a *sensor-to-sensor conversion* scheme. To achieve this, we leverage the output high-resolution Gaussians from our model and ray-trace the LiDAR rays to obtain the corresponding distances. Thanks to our clean voxel scaffold, the reconstructed scene is free of floaters and we set the opacity $\alpha$ to 1 for all the Gaussians to ensure a *hard* intersection that aligns better with the geometry.

**Application: Text-to-Scene Generation.** Our method can be easily extended to generate 3D scenes from text prompts. Similar to MVDream [47], we train a multi-view diffusion model with the architecture of VideoLDM [2] that generates images from text prompts. The original spatial self-attention layer is inflated along the view dimension to achieve content consistency [25, 65]. For training, we use CogVLM [58] to annotate the images automatically on a large scale. After the model is trained, we directly feed the output of the multi-view model to SCube to lift the observations into 3D space for novel view synthesis.

## 4 Experiments

In this section, we validate the effectiveness of SCube. First, we present our new data curation pipeline that produces ground-truth voxel grids (§ 4.1). Next, we demonstrate SCube's capabilities in scene reconstruction (§ 4.2), and further highlight its usefulness in assisting the state-of-the-art Gaussian splatting pipeline (§ 4.3). Finally, we showcase other applications of our method (§ 4.4) and perform ablation studies to justify our design choices (§ 4.5).

### 4.1 Dataset Processing

Accurate 3D data is essential for our method to learn useful geometry and appearance priors. Fortunately, many autonomous driving datasets [3, 53] are equipped with 3D LiDAR data, and one can simply accumulate the point clouds to obtain the 3D scene geometry [20, 40]. However, the LiDAR

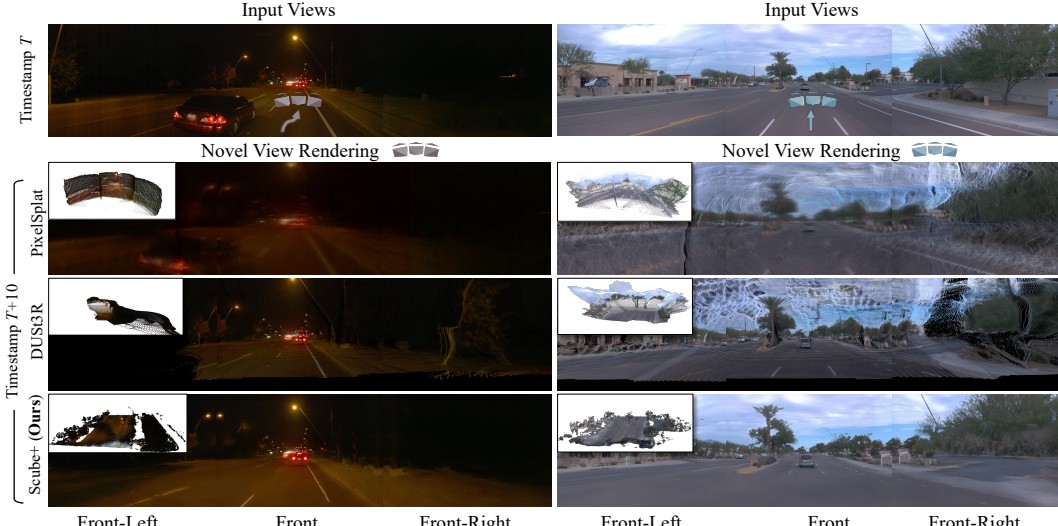

Figure 4: **Novel View Synthesis.** We show the synthesized novel views of SCube+ compared to baselines approaches. The inset of each subfigure shows a top-down visualization (an extreme novel view) of the reconstructed scene geometry.

points usually do not cover high-up regions such as tall buildings and contain dynamic (non-rigid) objects that are non-trivial to accumulate.

We hence build a data processing pipeline based on Waymo Open Dataset [53] as shown in Fig. 3, consisting of three steps: **Step 1**, we accumulate the LiDAR points in the world space, removing the points within the bounding boxes of dynamic objects such as cars and pedestrians. We additionally obtain the semantics of each accumulated LiDAR point, where non-annotated points are assigned the semantics of their nearest annotated neighbors. **Step 2**, we use the multi-view stereo (MVS) algorithm available in COLMAP [44] to reconstruct the dense 3D point cloud from the images, and the semantic labels of the points are obtained by Segformer [64]. **Step 3**, we add point samples for the dynamic objects according to their bounding boxes at a given target frame. This gives us

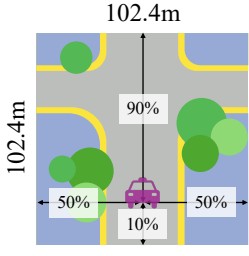

static *and* accumulated ground truths available for training. For each data sample, we crop the point cloud into a local chunk of $102.4m \times 102.4m$ centered around a randomly sampled ego-vehicle pose. Since there are no rear-view cameras in the Waymo dataset, we allocate more space for the chunk in the forward direction, as shown in the inset figure. See Appendix A for additional details.

## 4.2 Large-scale Scene Reconstruction

**Evaluation and Baselines.** To assess our method's power for 3D scene reconstruction, we follow the common protocol to evaluate the task of novel view synthesis [4, 42, 68]. Given input multi-view images (details about choosing views are in Appendix A) at frame $T$, we render novel views at future timestamps $T + 5$ and $T + 10$, and compare them to the corresponding ground-truth frames by calculating Peak Signal-to-Noise Ratio (PSNR), Structural Similarity Index Measure (SSIM), and Learned Perceptual Image Patch Similarity (LPIPS) [74]. We exclude the regions of moving objects for $T + 5$ and $T + 10$ evaluation, and only use three front views when computing the metrics.

We use PixelNeRF [68], PixelSplat [4], DUSt3R [57], MVSplat [6], and MVSGaussian [29] as our baselines for comparison. [4, 6, 29, 68] take images and their corresponding camera parameters as input and reconstruct NeRFs or 3D Gaussian representations. DUSt3R [57] directly estimates per-pixel point clouds from the images. We append additional heads to the its decoder which predicts other 3D Gaussian attributes along with the mean positions and finetune it with a rendering loss. For all other baselines, we take the official code and re-train them on our dataset. We tried to add the

|  | Reconstruction ($T$) | | | Prediction ($T+5$) | | | Prediction ($T+10$) | | |
|---|---|---|---|---|---|---|---|---|---|
|  | PSNR↑ | SSIM↑ | LPIPS↓ | PSNR↑ | SSIM↑ | LPIPS↓ | PSNR↑ | SSIM↑ | LPIPS↓ |
| PixelNeRF [68] | 15.26 | 0.51 | 0.66 | 15.21 | 0.52 | 0.64 | 14.61 | 0.49 | 0.66 |
| PixelSplat [4] | 22.15 | 0.71 | 0.61 | 20.11 | 0.70 | 0.60 | 18.77 | 0.66 | 0.62 |
| DUSt3R [57] | 17.17 | 0.60 | 0.58 | 17.08 | 0.62 | 0.56 | 16.08 | 0.58 | 0.60 |
| MVSplat [6] | 21.84 | 0.71 | 0.46 | 20.14 | 0.71 | 0.48 | 18.78 | 0.69 | 0.52 |
| MVSGaussian [29] | 21.25 | 0.80 | 0.51 | 16.49 | 0.70 | 0.60 | 16.42 | 0.60 | 0.59 |
| SCube (**Ours**) | 25.90 | 0.77 | 0.45 | 19.90 | 0.72 | 0.47 | 18.78 | 0.70 | 0.49 |
| SCube+ (**Ours**) | **28.01** | **0.81** | **0.25** | **22.32** | **0.74** | **0.34** | **21.09** | **0.72** | **0.38** |

Table 1: **Quantitative Comparisons on 3D Reconstruction.** The metrics are computed both at the input frame $T$ and future frames. ↑: higher is better, ↓: lower is better.

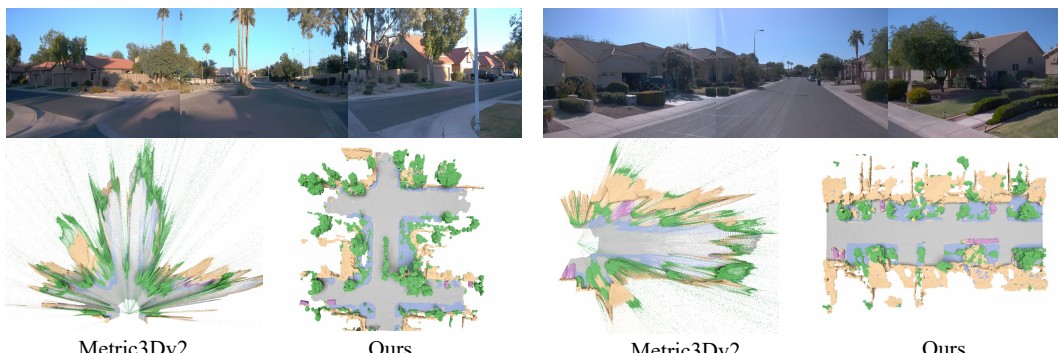

| Metric3Dv2 | Ours | Metric3Dv2 | Ours |

Figure 5: **Geometry Reconstruction from Sparse Views.** We show the comparison between our method and Metric3Dv2 [18]. The semantics of Metric3Dv2 are obtained from Segformer [64].

state-of-the-art depth estimator Metric3Dv2 [18] for depth supervision but empirically found that the performance degraded.

**Results and Analysis.** We show our results quantitatively in Tab. 1 and visually in Fig. 4. Our method outperforms all the baselines for both the current frame (reconstruction) and future frames (prediction) by a considerable margin on all metrics. PixelNeRF is limited by the representation power of the network, and simply fails to capture high-frequency details in the scene. PixelSplat highly relies on overlapping regions in the input views and cannot adapt well to our sparse view setting. It fails to model the true 3D geometry as shown in the top-down view, and simply collapses the images into constant depths. The multi-view-stereo-based methods [6, 29] cannot enable extreme novel view synthesis such as the bird-eye view, and could not recover highly-occluded regions. Thanks to the effective pre-training of DUSt3R, it is able to learn plausible displacements in the image domain, but the method still suffers from missing regions, misalignments, or inaccurate depth boundaries. In contrast, our method can reconstruct complete scene geometry even for far-away regions. It is both accurate and consistent while producing high-quality novel view renderings.

To better demonstrate the power of learning priors in 3D, we build another baseline using the state-of-the-art metric depth estimator Metric3Dv2 [18] to unproject the images into point clouds using 2D learned priors. As shown in Fig. 5, our method can reconstruct more complete, uniform, and accurate scenes, justifying the power of representing and learning geometry directly in the true 3D space.

### 4.3 Assisting Gaussian Splatting Initialization

Our method creates scene-level 3D Gaussians with accurate geometry and appearance, which can be used to initialize large-scale 3D Gaussian splatting [23] training. This is particularly useful in outdoor driving scenarios where structure-from-motion (SfM) may fail due to the sparsity of viewpoints.

To demonstrate this, we consider and compare three initialization methods: **Random** initialization is where points are uniformly sampled within the range of $(-20\text{m}, 20\text{m})^3$ around each camera. **Metric3Dv2** initialization is where we use the unprojected cloud from Metric3Dv2 [18]'s monocular

| | $R = 10$ | | | $R = 20$ | | | $R = 40$ | | |
|---|---|---|---|---|---|---|---|---|---|
| | PSNR↑ | SSIM↑ | LPIPS↓ | PSNR↑ | SSIM↑ | LPIPS↓ | PSNR↑ | SSIM↑ | LPIPS↓ |
| Random | 21.66 | 0.72 | 0.38 | 24.27 | 0.78 | 0.34 | 24.93 | 0.80 | 0.35 |
| Metric3Dv2 [18] | 23.30 | 0.75 | 0.33 | 25.21 | 0.80 | 0.32 | 25.58 | 0.80 | 0.34 |
| SCube (**Ours**) | **24.10** | **0.77** | **0.32** | **25.94** | **0.81** | **0.30** | **26.07** | **0.82** | **0.32** |

Table 2: **Initializations for Gaussian Splatting training.** We train 3D Gaussians with different initialization for $R$ frames. We report the test-set metrics. ↑: higher is better, ↓: lower is better.

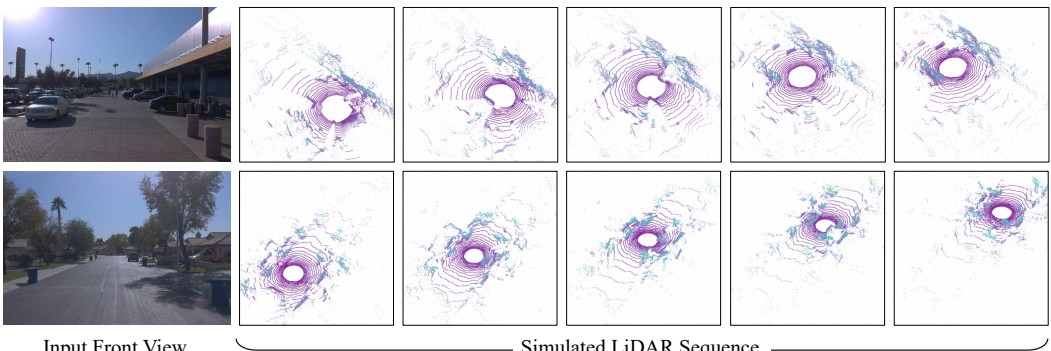

Input Front View — Simulated LiDAR Sequence

Figure 6: **LiDAR Simulation.** We demonstrate qualitative results of image-to-consistent-LiDAR transfer. The LiDAR sequences are simulated by moving the camera forward by 60 meters.

depth and align its scale to metric-scale LiDAR. **SCube (ours)** initialization directly adopts the positions and colors of the Gaussians from our pipeline. For input to these methods, we choose the views from the first frame $T$ and control the number of initial points to 200k. We then incorporate $R$ subsequent frames into the full training, with every 3 frames skipped to be used in the test set. The number of training iterations is fixed to 30k and the initial positional learning rate is set to $1.6e-5$. We select 15 static scenes for experiments and report their average metrics, which are shown in Tab. 2. The results consistently demonstrate SCube's effectiveness as an initialization strategy that provides accurate 3D grounding and alleviates overfitting on the training views.

### 4.4 Other Applications

We demonstrate the applications of our method as described in § 3.3. Fig. 6 shows the consistent LiDAR simulation results, where the simulated sequences could effectively cover a long range away from the input camera positions, while resolving intricate geometric structures such as buildings, trees, or poles. Fig. 7 exemplifies the text-to-scene generation capability enabled by our method. The 3D geometry and appearance respect the input text prompt and the corresponding images. Readers are referred to Appendix D.3 for more generative results.

### 4.5 Ablation Study

**Image-Conditioning Strategy.** We replace the image conditioning strategy described in Eq (1) in the voxel grid reconstruction stage with a vanilla scheme that broadcasts the same feature to all the voxels along the pixel's ray. The final IoU of the fine-level voxel grid drops from $34.31\%$ to $30.33\%$, and the mIoU that considers the accuracy of the voxel's semantic prediction drops from $20.00\%$ to $16.61\%$. This shows the effectiveness of our conditioning strategy being able to disambiguate voxels at different depths.

**Two-stage Reconstruction.** We disentangle the voxel grid and appearance reconstruction stages to make the best use of different types of models. Using a single-stage model[2] that simultaneously

---

[2]In practice, we test the upper bound of the single-stage model by feeding in ground-truth $1024^3$ voxel grids, because otherwise the fully-dense high-resolution condition will lead to out-of-memory.

*... a palm tree, a road with traffic, lined with trees and buildings, under a blue sky with scattered white clouds, ...*

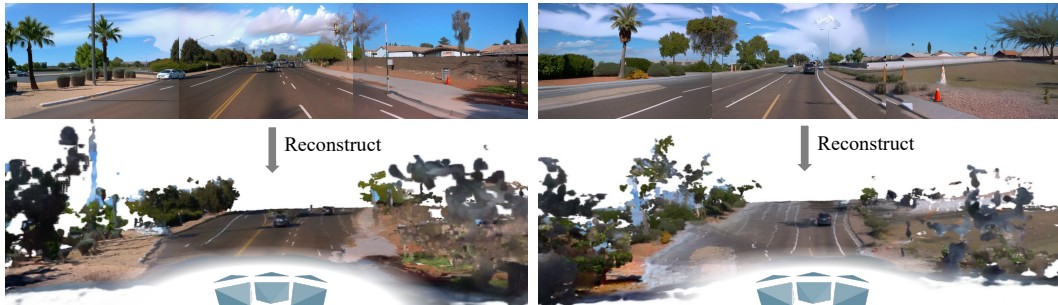

Figure 7: **Text-2-Scene Generation.** Given a text prompt, we could generate various multi-view images and lift them to 3D scenes with SCube. See Appendix D.3 for more text-2-scene results.

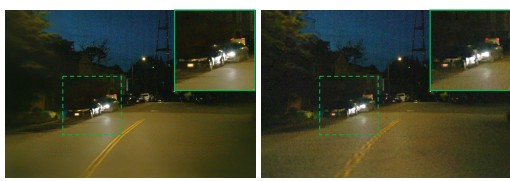

Figure 8: **Effects of GAN Postprocessing.** Left: SCube+; Right: SCube.

| Resolution | $M$ | PSNR↑ | LPIPS↓ |
|---|---|---|---|
| $256^3$ | 4 | 18.58 | 0.62 |
| $1024^3$ | 1 | 19.34 | 0.52 |
| $1024^3$ | 4 | 19.34 | 0.48 |

Table 3: **Ablation Study for Appearance Reconstruction.**

predicts the sparse voxels and the appearance from images, we can only achieve a PSNR/LPIPS of $17.88/0.57$, in comparison to $19.34/0.48$ when using the two-stage model. Here the values are the average of $T+5$ and $T+10$ frames. In terms of geometry quality, the single-stage model is also significantly worse (up to $100\times$) in Chamfer distance than the two-stage model. Please refer to more details about the analysis of geometry quality in Appendix D.1.

**Appearance Reconstruction.** We validate the effect of voxel grid resolution and the number of Gaussians per voxel $M$ in the appearance reconstruction stage. Results are shown in Tab. 3. We find that higher-resolution voxel grids are crucial for capturing detailed geometry, and using a larger number of Gaussians only slightly increases the performance. Thus, we set $M=4$ as a moderate value for the final results. Compared in Fig. 8, the GAN-based postprocessing, despite the time cost, is beneficial for producing high-quality images by sharpening the renderings. See Appendix D.2 for more visual comparisons.

## 5 Discussion

**Conclusion.** In this work, we have introduced SCube, a feed-forward method for large 3D scene reconstruction from images. Given sparse view non-overlapping images, our method is able to predict a high-resolution 3D scene representation consisting of voxel-supported Gaussian splats (VoxSplat) and a light-weight sky panorama in a single forward pass within tens of seconds. We have demonstrated the effectiveness of our method on the Waymo Open Dataset, and have shown that our method outperforms the state-of-the-art methods in terms of reconstruction quality.

**Limitations.** Our method does suffer from some limitations. First, the current method is not able to handle complicated scenarios such as dynamic scenes under extreme lighting or weather conditions. Second, the quality of appearance in occluded regions still carries uncertainty. Third, the method itself still requires ground-truth 3D training data which is not always available for generic outdoor scenes. In future work, we plan to address these limitations by incorporating more advanced neural rendering techniques and by exploring more effective ways to generate training data.

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

**– Appendix –**

# A  Implementation Details

**Additional Data Processing Details.**  For each data sample, we crop the point cloud obtained from § 4.1 into a local chunk of $102.4\text{m} \times 102.4\text{m}$. The point cloud is then voxelized into the fine-level and coarse-level grids used in § 3.1 with $1024^3$ and $256^3$ resolutions respectively (with voxel sizes of 0.1m and 0.4m). Our dataset contains 20243 chunks for training and 5380 chunks for evaluation, out of the 798 training and 202 validation sequences.

**Input and Evaluation Details.** Waymo dataset provides 5 views for each camera frame, namely `front`, `front-left`, `front-right`, `side-left` and `side-right`. However, not all of the baseline methods we compared with in § 4.2 can handle the unconventional camera intrinsic in the `side-left` and `side-right` views. We hence only use the first three views (with a resolution of $1920 \times 1280$) in § 4.2 for both the input and the evaluation metrics. However, in § 4.3 we opt to use all 5 views for the input to both our method and the baseline due to compatibility and maximized performance.

For the baselines, the original PixelSplat [4] method does not have depth supervision. To make the comparison fair, we attempt to add a depth supervision loss to it. However, the experimental result shows that the additional loss hurts the performance as shown in Tab. 4. We thus report the results of vanilla PixelSplat in the main paper.

|  | $T+5$ | | | $T+10$ | | |
|---|---|---|---|---|---|---|
|  | PSNR↑ | SSIM↑ | LPIPS↓ | PSNR↑ | SSIM↑ | LPIPS↓ |
| PixelSplat [4] | **20.11** | **0.70** | **0.60** | 18.77 | **0.66** | **0.62** |
| PixelSplat [4] w/ Depth Supervision | 19.91 | 0.58 | 0.66 | **18.87** | 0.56 | 0.67 |

Table 4: **Comparison of PixelSplat and PixelSplat with Depth Supervision.**

**Training Details.** The diffusion loss in Eq (2) is defined similar to [16, 40] with a $v$-parametrization as:

$$\mathcal{L}_{\text{Diffusion}} = \mathbb{E}_{t,\mathbf{X},\boldsymbol{\epsilon}\sim\mathcal{N}(0,I)} \left[ \left\| \boldsymbol{v}(\sqrt{\bar{\alpha}_t}\mathbf{X} + \sqrt{1-\bar{\alpha}_t}\boldsymbol{\epsilon}, t) - (\sqrt{\bar{\alpha}_t}\boldsymbol{\epsilon} - \sqrt{1-\bar{\alpha}_t}\mathbf{X}) \right\|_2^2 \right], \qquad (7)$$

where $\boldsymbol{v}(\cdot)$ is the diffusion network, $t$ is the randomly sampled diffusion timestamp, and $\bar{\alpha}_t$ is the scheduling factor for the diffusion process, whose details are referred to in [16].

We train all of our models using the Adam [24] optimizer with $\beta_1 = 0.9$ and $\beta_1 = 0.999$. We use PyTorch Lightning [10] for building our distributed training framework. For the voxel grid reconstruction stage, we train both coarse-level and fine-level voxel latent diffusion models with $64\times$ NVIDIA Tesla A100s for 2 days. For the appearance reconstruction model, we train it using $8\times$ NVIDIA Tesla A100s for 2 days. Empirically, we use $\lambda = 1.0$ for $\mathcal{L}_{\text{Depth}}$ in Eq (2). Additionally, we use $\lambda_1 = 0.9$, $\lambda_2 = 1.0$, $\lambda_{\text{SSIM}} = 0.1$ and $\lambda_{\text{LPIPS}} = 0.6$ in Eq (6). For image condition, we set the feature channel $C = 32$, the number of depth bins $D = 64$, $z_{\text{near}} = 0.1$ and $z_{\text{far}} = 90.0$. We linearly increase the interval of depth bins.

# B  Network Architecture

**Voxel Grid Reconstruction.**   We follow [40] to implement the Sparse Structure VAE and the Diffusion UNet using the sparse 3D deep learning framework fVDB [62]. Hyperparameters for training them are listed in Tab. 5 and Tab. 6. We pass the images to distilled DINO-v2 [34] ViT-B/14. We use four 2D convolutional layers (channel dims: $[768, 256, 256, 32, 32]$, kernel size: 3, stride: 1) to further process the DINO-v2 output to predict the image feature and the depth distribution.

**Appearance Reconstruction.**  We process the original input images with three 2D convolutional layers (channel dims: $[3, 16, 32, 32]$, kernel size: 3, stride: $[1, 1, 2]$). For the last two convolutional

|  | Waymo $64^3 \to 256^3$ | Waymo $256^3 \to 1024^3$ |
|---|---|---|
| Model Size | 14.9M | 3.8M |
| Base Channels | 64 | 32 |
| Channels Multiple | 1,2,4 | 1,2,4 |
| Latent Dim | 8 | 8 |
| Batch Size | 32 | 32 |
| Epochs | 50 | 50 |
| Learning Rate | | 1e-4 |

Table 5: **Hyperparameters for VAE.**

|  | Waymo - $64^3$ | Waymo - $256^3$ |
|---|---|---|
| Diffusion Steps | | 1000 |
| Noise Schedule | | linear |
| Model Size | 728M | 83.0M |
| Base Channels | 192 | 64 |
| Depth | | 2 |
| Channels Multiple | 1,2,4,4 | 1,2,2,4 |
| Heads | | 8 |
| Attention Resolution | 16 | 32 |
| Dropout | 0.0 | 0.0 |
| Batch Size | 512 | 256 |
| Iterations | 40K | 20K |
| Learning Rate | | 5e-5 |

Table 6: **Hyperparameters for voxel latent diffusion models.**

layers, we set the residual connections. We additionally positionally encode each voxel and then concatenate the positional encoding [32] of each voxel with the corresponding voxel feature after ray casting. We then apply a 3D sparse UNet to output per-Gaussian parameters. We use GT voxels in appearance reconstruction training. Hyperparameters of this 3D sparse UNet are listed Tab. 7.

| Model Size | Base Channels | Channels Multiple | Batch Size | Epochs | Learning Rate |
|---|---|---|---|---|---|
| 4.3M | 32 | [1, 2, 4] | 32 | 15 | 1e-4 |

Table 7: **Hyperparameters for 3D sparse UNet in appearance reconstruction stage.**

**Sky Panorama for Background.** For the sky panorama model, we set $H_p = 768, W_p = 1536$ in the training stage and increase $H_p = 1024, W_p = 2048$ in the inference time. To decode sampled sky features into the RGB image, we utilize a 2D CNN network reducing the channel from 32 to 16 to 3 with stride 1, keeping the spatial resolution unchanged.

## C  SCube+ without Per-scene Training

In § 3.3 we introduce a GAN postprocessing module to refine the rendered images, which is finetuned on each scene. To further improve the efficiency of our method, we hereby present a postprocessing module that is jointly trained on the full dataset, without the need of per-scene finetuning. Specifically, we replace the original GAN with a pix2pix-turbo model [35] (which we denote as SCube+*) and train it with image pairs inferred from our model and the ground truths. The results are shown in Fig. 9. This improved model not only reduces the voxel block artifacts, but also resolve the ISP inconsistencies within the image. After enabling this module, the FPS drops from 138 to 20 but can still be visualized interactively.

| | Reconstruction ($T$) | | | Prediction ($T + 5$) | | | Prediction ($T + 10$) | | |
|---|---|---|---|---|---|---|---|---|---|
| | PSNR↑ | SSIM↑ | LPIPS↓ | PSNR↑ | SSIM↑ | LPIPS↓ | PSNR↑ | SSIM↑ | LPIPS↓ |
| SCube | 25.90 | 0.77 | 0.45 | 19.90 | 0.72 | 0.47 | 18.78 | 0.70 | 0.49 |
| SCube+ | **28.01** | **0.81** | **0.25** | **22.32** | **0.74** | **0.34** | **21.09** | **0.72** | **0.38** |
| SCube+* | 22.59 | 0.68 | 0.38 | 20.37 | 0.66 | 0.41 | 19.65 | 0.65 | 0.42 |

Table 8: **Quantitative Comparisons on 3D Reconstruction.** The metrics are computed both at the input frame $T$ and future frames. ↑: higher is better, ↓: lower is better.

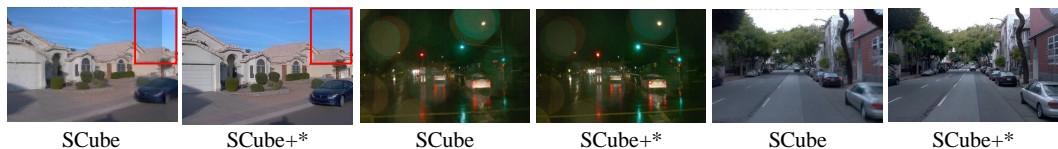

| SCube | SCube+* | SCube | SCube+* | SCube | SCube+* |

Figure 9: **SCube+*.** Results from the postprocessing network without per-scene optimization. White-balance inconsistencies from different views (marked in red box) can be fixed.

# D  Additional Results

In this section, we provide more qualitative results on all datasets. We additionally provide a **supplementary video** in the accompanying files to better illustrate our results.

## D.1  Geometry Quality

We note that the uncertainty of the scene geometry given our input images is large, and the problem that the model tackles is indeed non-trivial and sometimes even ill-posed. To demonstrate this, we compute the percentage of occluded voxels (that are invisible from the input images) w.r.t. all the ground-truth voxels, and the number is around 80%. To quantitatively evaluate the geometry quality, we compute an additional metric called 'voxel Chamfer distance' that measures the L2-Chamfer distance between the predicted voxels and ground-truth voxels (that are pixel-aligned), divided by the voxel size. This metric reflects the geometric accuracy of our prediction by measuring on average how many voxels is the prediction apart from the ground truth. The results on Waymo Open Dataset are shown in Tab. 9.

| Quantile | 0.5 _(median)_ | 0.6 | 0.7 | 0.8 | 0.9 |
|---|---|---|---|---|---|
| **Ours** | 0.26 | 0.28 | 0.32 | 0.37 | 0.51 |

Table 9: **Geometry Quality Comparison.** We show the voxel Chamfer distance comparison between our two-stage model and a single-stage non-diffusion model.

Tab. 9 indicates that on 90% of the test samples, the predicted voxel grid is only half of a voxel off from the ground truth. We note that during our data curation process, there could be errors in the ground-truth voxels (_e.g._, due to COLMAP failures), accounting for the outliers in the above metric. In the meantime, we visualize the sample with the worst voxel Chamfer distance in Fig. 10. The predicted results are decent even though the ground truth is corrupted due to the lack of motion in the ego car. This demonstrates the robustness of our method.

## D.2  Visual Ablation Study

In addition to the quantitative ablation study in Tab. 3, we present a qualitative demonstration in Fig. 11. For the single-stage model, we test the upper bound of it by feeding the ground-truth $1024^3$ voxel grids because otherwise the fully-dense high-resolution condition will lead to out-of-memory. The qualitative results match the numbers, showing the importance of using higher-resolution voxel grids and the two-stage model.

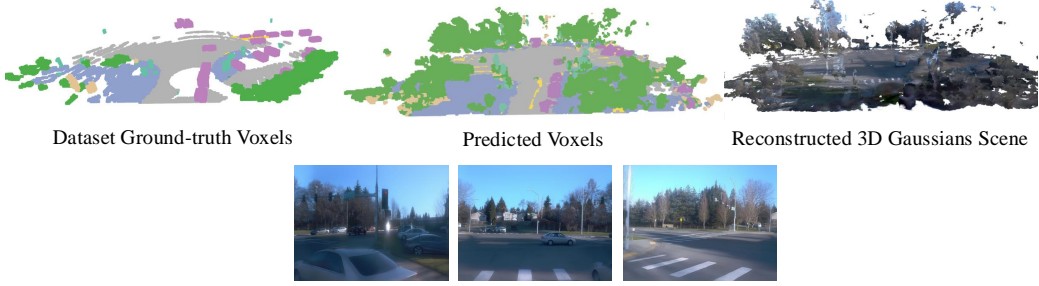

Dataset Ground-truth Voxels      Predicted Voxels      Reconstructed 3D Gaussians Scene

Final Rendering

Figure 10: Result on the data sample with the worst voxel Chamfer distance. We show geometry reconstruction and the image renderings.

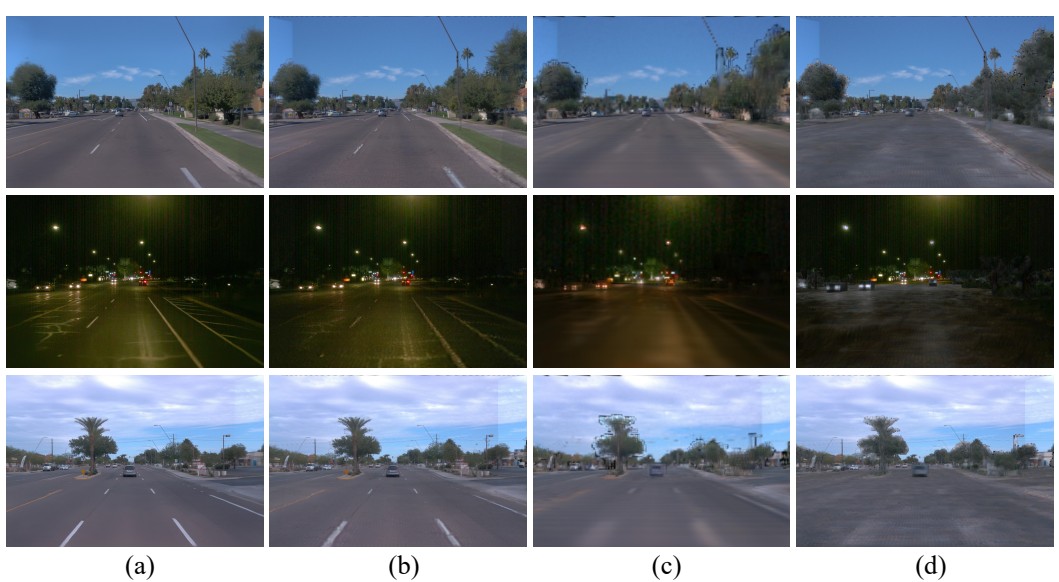

(a)      (b)      (c)      (d)

Figure 11: **Visual Ablation Study.** (a) SCube+ (b) SCube (c) SCube with a $256^3$ resolution input grid (d) Single-stage model. Zoom in for a better view.

## D.3 Additional Results on Text-2-Scene Generation

We provide additional text-2-scene generation results in Fig. 12 and Fig. 13.

*A residential neighborhood features houses with well-maintained gardens, autumn-colored trees, lawns with scattered leaves, parked cars, driveways, and clear blue skies.*

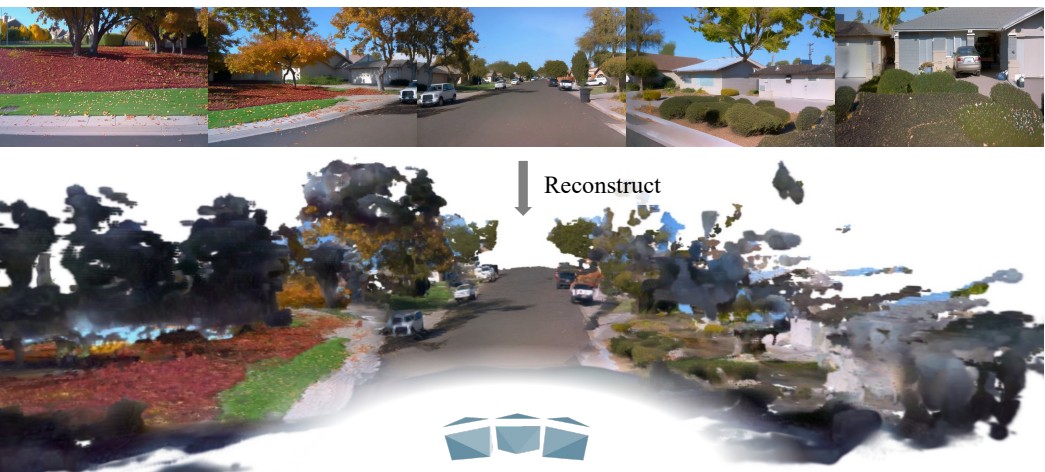

*A residential area features multiple houses, some with specific decorations and vehicles parked outside, including a white pickup truck and a gray car, along with various greenery and utility elements.*

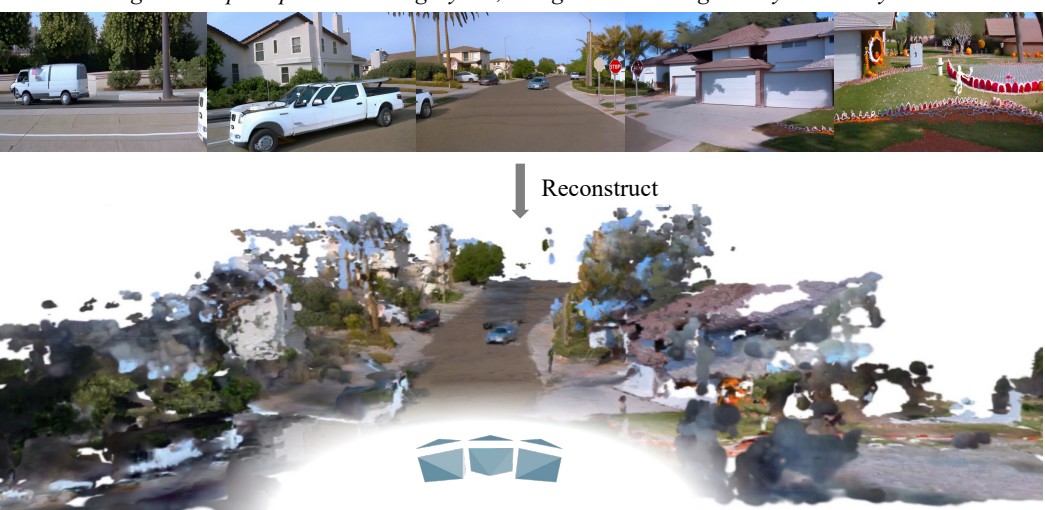

Figure 12: **More Text-2-Scene Generation.** The generated multi-view images may contain flaws, while SCube is still able to reconstruct the 3D scenes.

*A suburban neighborhood features two-story houses with reddish-brown roofs and beige walls, marked roads, various parked vehicles, stop signs, and a mixture of gravel, rocks, and trees providing shade on a sunny day.*

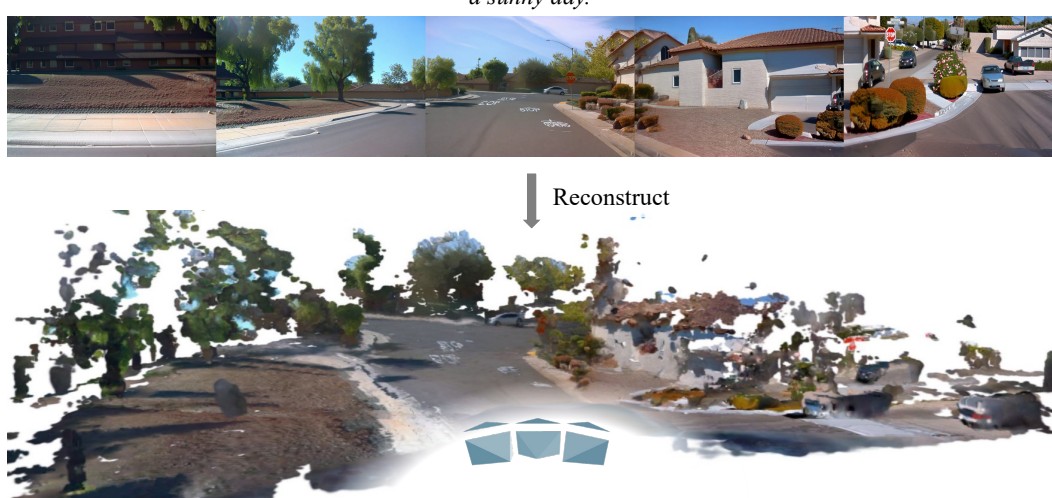

Reconstruct

*A suburban neighborhood features a park with green trees, residential houses with red-tiled roofs, streets with bike lane signs and white markings, well-maintained lawns, and sidewalks.*

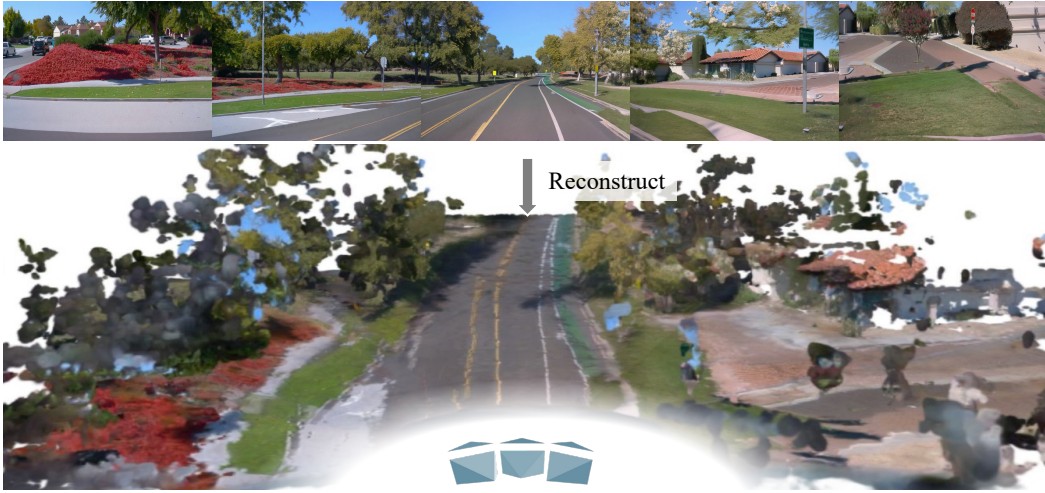

Reconstruct

Figure 13: **More Text-2-Scene Generation.**

## D.4   Additional Results on Large-scale Scene Reconstruction

We provide additional results on large-scale scene reconstruction from real-world captures in Fig. 14.

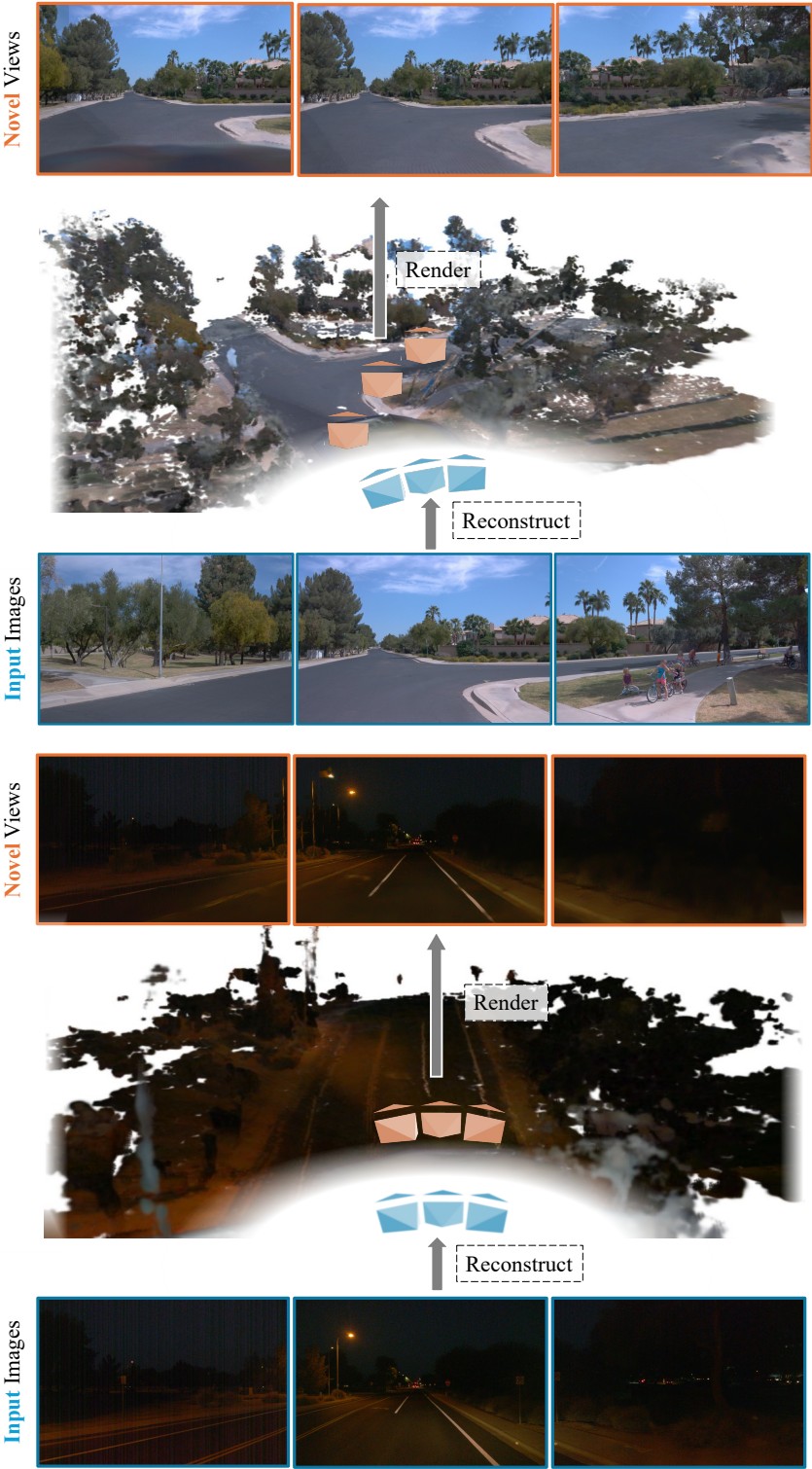

Figure 14: **More Novel View Synthesis.**  Our method is able to synthesis extreme novel views.

### D.5 Additional Results on LiDAR Simulation

We provide additional LiDAR Simulation results in Fig. 15. We also show the result on a long sequence input in Fig. 16.

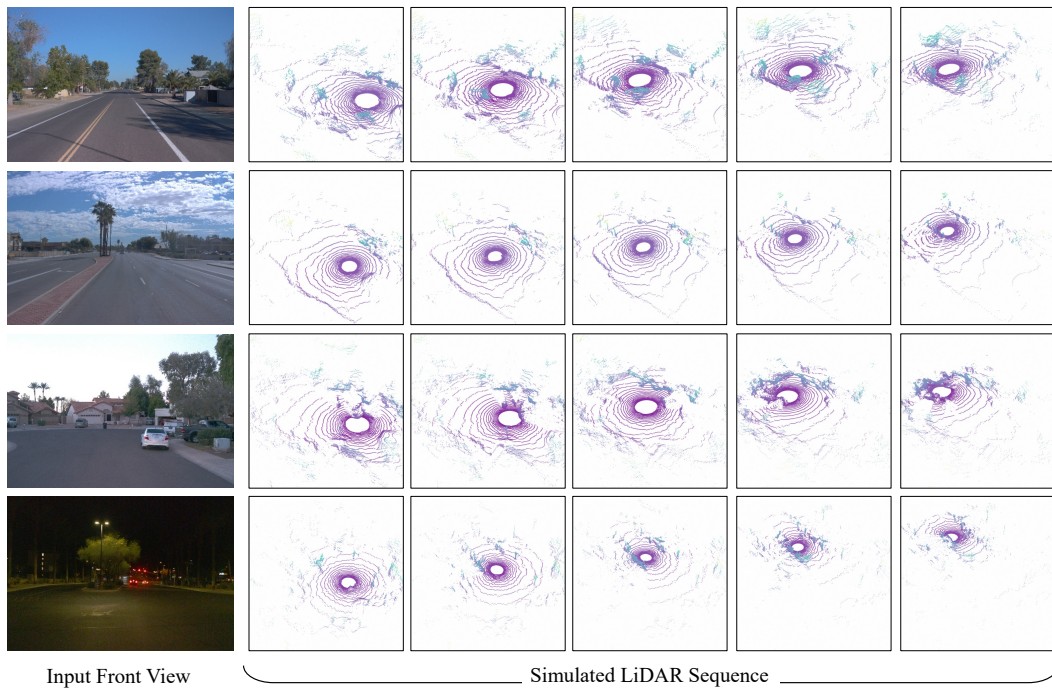

Input Front View           Simulated LiDAR Sequence

Figure 15: **More LiDAR Simulation results.**

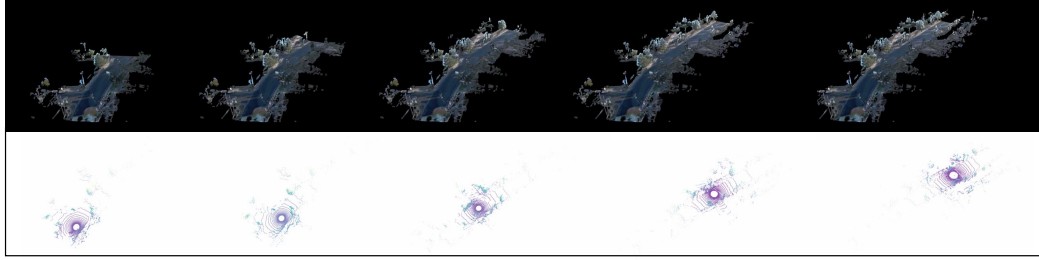

Figure 16: **SCube with Long Sequence Input.** Up: reconstructed scene with appearance. Down: LiDAR simulation result. We chunk the long sequence into clips and apply out method iteratively.

