# OpenReview forum: "SCube: Instant Large-Scale Scene Reconstruction using VoxSplats"
_NeurIPS.cc/2024/Conference — NeurIPS 2024 poster_

### Official Review · Reviewer_NKmV · 2024-07-12

**Soundness:** 3
**Presentation:** 3
**Contribution:** 2
**Rating:** 5
**Confidence:** 4

**Summary:**

This paper presents a method to reconstruct large-scale 3D scenes from a sparse set of posed images. The method has two stages: the first stage is for voxel grid reconstruction, which is based on the XCube [38], a 3D generative model, and the authors introduce the image condition to the model.  The second stage is for appearance reconstruction, which is a feedforward appearance prediction model to predict a certain number of Gaussians for each voxel. The authors also propose a GAN-based post-processing step to boost the appearance of the reconstructed scene. The proposed method is evaluated on the Waymo self-driving dataset, showing the superiority of the proposed method compared to prior art on 3D reconstruction tasks. Besides, the authors also demonstrate the applications of the proposed method, such as LiDAR simulation and text-to-scene generation.

**Strengths:**

- The proposed method addresses an important problem in 3D vision, i.e., reconstructing large-scale 3D scenes from a sparse set of posed images. The authors cleverly take advantage of the pre-trained 3D diffusion generative model to relieve the ill-posedness of the problem.
- Although technically the proposed method is not entirely new, the paper presents a framework to combine the voxel grid reconstruction and appearance prediction, and the presented result looks promising to me.
- The paper is well-written and easy to follow. The authors provide a clear explanation of the proposed method and the experiments.

**Weaknesses:**

- The proposed representation's abbreviation "VoxSplats" is somewhat confusing to me, since the term "splatting" in original 3DGS refers to the rendering technique that splats the 3D Gaussians onto the image plane, rather than the 3D representation. The proposed representation is more like a hybrid of voxel grids and Gaussian representation. The authors may consider using a more accurate term to describe the representation.
- It's unclear how the foreground and background are separated in the proposed method. Using a panorama to represent the sky makes sense as the sky is usually far away from the scene, but some buildings are treated as the background and seem to be represented by the panorama as well in the supplementary video. A more detailed explanation and discussion of the foreground-background separation is needed.
- There are some lighting/exposure/white-balance inconsistencies in the reconstructed scenes in the supplementary video. The authors should discuss the limitations of the proposed method in handling the lighting/exposure/white-balance variations in the input images.
- For SCube+, how to guarantee consistency among the rendered views? Also, rendering with the proposed GAN-based post-processing step is time-consuming, which would destroy the valuable real-time rendering performance of the Gaussian Splatting. The authors should discuss these issues in the paper.
- Missing the evaluation of the geometric accuracy of the reconstructed scenes. As the rendering quality of SCube is not so good, geometric accuracy is crucial to the applications of the reconstructed scenes. The authors should provide a detailed evaluation of the geometric accuracy of the reconstructed scenes.

**Questions:**

The most important questions that I would like the authors to address are:
- Geometric accuracy evaluation of the reconstructed scenes.

**Limitations:**

Yes, the authors have discussed the limitations of the proposed method.

---

> ### Author Rebuttal · Authors · 2024-08-07
>
> Thank you so much for your detailed review. We are glad to understand that you have found the results promising and the paper well-written. We hereby quote and answer the raised questions as below:
>
> > **Naming of VoxSplat.** The authors may consider using a more accurate term to describe the representation.
>
>  Thank you for the constructive feedback, and we are actively considering renaming our representation to, e.g., `VoxGaussian` for accuracy. We would still use the term `VoxSplat` in this rebuttal though to avoid confusion, but we will change the name in the revised version.
>
> > **Foreground-background separation.** It's unclear how the foreground and background are separated. Using a panorama to represent the sky makes sense as the sky is usually far away from the scene, but some buildings are treated as the background and seem to be represented by the panorama.
>
> Conceptually the scene to be reconstructed could be decomposed into three parts:
>
> 1. *Foreground*: This refers to the sparse voxel grids within the 102.4m region enclosing the camera (as visualized in Appendix A). This is the main part we reconstruct using our VoxSplat representation.
> 2. *Background*: This is the sky panorama that we explained in L160-165, which is assumed to be at infinity.
> 3. *Midground*: This is the region between *Foreground* and *Background* which, as the reviewer has pointed out, we do not explicitly tackle in our framework.
>
> For the novel view synthesis task where we render the sky panorama, buildings in the *Midground* are baked into the *Background*. We humbly argue that this is a sufficient approximation for this task at hand, since the novel viewpoint to be rendered is not drastically far away from the input views, so the buildings far away can be approximated in the panorama. For geometry reconstruction or bird-eye-view visualization purposes, the *Midground* as well as the *Background* are both cropped.
> Notably, since our ultimate goal is to build *true 3D* geometry, developing ad-hoc ways for modeling the midground is not a principled solution for us. To solve this in a principled manner, we choose to apply our framework to a longer sequence input (as shown in `Fig.A` in the rebuttal PDF), in which case the *Midground* for the beginning frames is just *Foreground* in the subsequent frames and could be modeled explicitly by the VoxSplat representation.
>
> > **Handling white-balance inconsistencies.** There are some lighting/exposure/white-balance inconsistencies in the reconstructed scenes in the supplementary video.
>
> We agree that this is not explicitly handled currently and will add more discussions to the limitations. In fact, we note that our image-space postprocessing module (L178-186, i.e., **SCube+**) can already mitigate this issue. Notably, we have improved this part of the pipeline to remove the need for per-scene training by introducing a new postprocessing renderer that jointly learns on *all* the dataset samples. Similarly, we take the rendered image (with potential artifacts) and the ground-truth images as the training pairs, and find that this simple module could already resolve the ISP inconsistencies within the image. Please refer to `Fig.B` in the rebuttal PDF for a visualization.
>
> > **Rendering in SCube+.** For SCube+, how to guarantee consistency among the rendered views? Also, rendering with the proposed GAN-based post-processing step is time-consuming.
>
> The GAN post-processing module in SCube+ only aims to remove the quantization artifacts and refine the details (ref. Fig. 8), but will leave the general scene structure untouched. This means that our renderings are still *grounded/supported* by the underlying VoxSplat representation that is guaranteed to be consistent. Empirically we observe the recovered fine details in SCube+ are also decently consistent as shown in the supplementary video (e.g. 00:27 lower-right). We agree that enforcing consistency in a more principled way is meaningful future work and note that one may apply a video model instead to enforce temporal consistency.
>
> After enabling this module, the FPS drops from 138 to 20 but can still be visualized interactively. We will discuss this in our limitation and aim to improve the timing in future work. Please note that this step is only needed if higher-quality rendering is required, and one can alternatively choose to finetune the 3DGS representation with optimization as we have demonstrated in $\S$ 4.3.
>
> > **Evaluation of geometric accuracy.** geometric accuracy is crucial to the applications of the reconstructed scenes. The authors should provide a detailed evaluation of the geometric accuracy of the reconstructed scenes.
>
> Thank you for the suggestion. We already have a light measurement of voxel IoU at L278-283 in terms of semantic voxel accuracy and we will include a more detailed evaluation of the geometric accuracy in the revision.
> Specifically, we compute an additional metric called 'voxel Chamfer distance' that measures the L2 Chamfer distance between predicted voxels and ground-truth voxels, divided by the voxel size. This metric reflects the geometric accuracy of our prediction by measuring on average how many voxels is the prediction apart from the ground truth. The results on Waymo Open Dataset are as follows:
>
>
> | Quantile               | 0.5 (median) | 0.6  | 0.7  | 0.8  | 0.9  |
> | ---------------------- | ------------ | ---- | ---- | ---- | ---- |
> | Voxel Chamfer distance | **0.26**     | 0.28 | 0.32 | 0.37 | 0.51 |
>
> This reflects that the geometry is very accurate in comparison to the ground truth, on average being smaller than half of a voxel for more than 90% of the data samples.

---

### Official Review · Reviewer_PK9U · 2024-07-12

**Soundness:** 3
**Presentation:** 4
**Contribution:** 3
**Rating:** 7
**Confidence:** 4

**Summary:**

- The paper proposes a new method for sparse-view 3D reconstruction using 3DGS.
- The framework uses two stages:
    1) it learns a latent voxel grid (based on XCube) to represent the geometry.
    2) it trains an appearance model to decode the latent voxel grid into a set of Gaussians
- The method further uses a background model to handle the sky.
- The authors propose an (optional) GAN postprocessing which improves the visual quality at the cost of longer (20m) per-scene optimization.
- The method was evaluated on the Waymo Open Dataset, and it outperforms compared methods.

**Strengths:**

- The paper is very well written and easy to follow.
- The method is novel and the results look good.
- I really like the idea of having probabilistic latent space representation (VAE).
- The paper contains an ablation study illustrating the tradeoff between model complexity and performance as various hyperparameters are changed.

**Weaknesses:**

- Section Method "Training and Inference" is not very clear. It would be preferable to expand the diffusion loss from Appendix A, and clearly describe the input/targets.
- I would also like to have more details on the Sky Panorama model.
- The method was evaluated on a single dataset. It would be interesting to see how the method performs on other datasets.

**Questions:**

- How does the method compare to other methods like MVSplatting, MVSGaussians? How does it differ? How do you think it would compare in terms of performance?

**Limitations:**

- Limitations are properly discussed in the paper.

---

> ### Author Rebuttal · Authors · 2024-08-07
>
> Thank you for your encouraging feedback for SCube, we are glad to understand that our method is easy to follow and you enjoy the idea and the results of our paper. In the following text, we will try to address your concerns.
>
> > **Details about the diffusion loss.** Section "Training and Inference" is not very clear. It would be preferable to expand the diffusion loss from Appendix A, and clearly describe the input/targets.
>
> Thank you for your suggestion, and we will expand the descriptions and clarify the input and targets in the revised paper.
> The training loss $\mathcal{L}_\text{diffusion}$ (in Eq. 7) is derived by maximizing the log-likelihood of the data $\mathbf{X}$ distribution. It can also be viewed as a score-matching process whose loss encourages the alignment of the predicted stochastic score and the ground truth. In each training iteration, we sample a random variable $\mathbf{X}$ (in our case this is the sparse voxel hierarchy) and pass its noised version to the sparse U-Net, which aims to predict the added Gaussian noise ${\epsilon}$. Practically we re-parametrize the network with the v-parametrization technique as introduced in [a].
>
> > **Details about sky panorama model.** I would also like to have more details on the Sky Panorama model.
>
> Modeling the sky independently plays an important role in outdoor scene reconstruction [b], which helps the model to disentangle the foreground and the sky at infinite depth. We adopt a 2D panorama $\mathbf{L} \in R^{H\times W \times C}$ as the sky representation, and the sky modeling consists of the following two steps:
>
> 1) *Encode the 2D sky panorama*: Since the 2D panorama is an expanded (hemi-)sphere with inverse equirectangular projection, we can get a cartesian coordinate $P=(x,y,z)$ on a unit sphere for each pixel $(u, v)$ in the panorama $\mathbf{L}$. We then project $P$ to the image plane to retrieve the image feature (note that we zero the translation part of the camera pose since only the view direction decides the sky color). We also leverage the sky mask which is computed as the inverse of the rendered sparse voxel grid mask to make sure only sky areas are retrieved. After this step, sky features are stored in $\mathbf{L}$.
> 2) *Sample and decode features from the panorama*: Given a camera with image resolution $(H', W')$, intrinsics $K$ and camera pose $\xi$, we generate camera rays and calculate their hit point on the unit sphere. With equirectangular projection, we obtain their pixel positions on the 2D sky panorama and query its features via trilinear interpolation, resulting in a 2D feature map $\mathbf{F} \in {R}^{H'\times W' \times C}$. We finally decode it with a 2D CNN network to get the RGB sky image of $\mathbf{I} \in {R}^{H'\times W' \times 3}$, which is alpha-composited with the foreground.
>
> We use $H=1024, W=2048, C=32$ for the 2D sky panorama in our implementation. In the sampling and decoding step, the 2D CNN network includes several Convolution--BatchNorm--ReLU layers, reducing the channel number from 32 to 16 to 3 with stride 1. We will make the descriptions utmost clear in our revision.
>
> > **Results on other datasets.** The method was evaluated on a single dataset. It would be interesting to see how the method performs on other datasets.
>
> Our method is able to generalize well to novel inputs, as already demonstrated in the Text-to-Scene generation experiment (Fig. 7), where the images are taken from a multi-view diffusion generative model. Furthermore, we demonstrate the general applicability of our method on a new larger-scale internal dataset consisting of 1.1k clips. Each clip contains about 200 frames and has dense point clouds for the GT voxel generation. We show a reconstruction and novel view synthesis sample on this dataset in `Fig. C` in the PDF file. The result shows that our model is capable of learning over a larger dataset and obtaining better reconstructions. Notably, in this example, we demonstrate the flexibility of our model that also supports back-view inputs (REAR-LEFT and REAR-RIGHT) for a 360-degree novel view rendering.
>
>
> > **Comparison to MVSplat and MVSGaussians.** How does the method compare to other methods like MVSplatting, MVSGaussians? How does it differ? How do you think it would compare in terms of performance?
>
> The main difference of our model compared to multi-view stereo-based methods such as MVSplat or MVSGaussians is the usage of true 3D priors. Specifically, MVSplat uses cross-attention to learn image-space correspondences and infer pixel-aligned Gaussians, and MVSGaussians lift from the images to a cost volume that is strongly correlated to the target view for rendering. While these methods achieve good rendering quality, they cannot learn the distribution of the 3D geometry. Comparably:
> - Quality-wise, MVS-based method cannot enable *extreme* novel view synthesis such as the bird-eye view of the scene as shown in Fig. 1 and the supplementary video. They also cannot recover *highly-occluded regions* as shown in Fig. 5.
> - Performance-wise, with the support of the fast sparse convolution infrastructure and the Gaussian renderer, our reconstruction is built within seconds and rendered in real-time. This is comparable to the baselines.
>
> Thinking ahead, the availability of the 3D prior and latent spaces could allow more explicit 3D editing or control capability (e.g. for the traffic) in future works.
>
>
> ***Reference:***
>
> [a] Salimans et al. Progressive Distillation for Fast Sampling of Diffusion Models. ICLR 2022.
>
> [b] Wang et al. Neural Fields meet Explicit Geometric Representations for Inverse Rendering of Urban Scenes. CVPR 2023.

---

> > ### Comment · Reviewer_PK9U · 2024-08-13
> >
> > Thank you for the rebuttal and the nice explanation of the diffusion loss and the panorama model. I feel like including them in the paper will improve the quality of the method section. As for my main question, I don't feel like it has been properly addressed. I would have liked a quantitative comparison (even limited) with the suggested baselines. While I still like the paper, I've decided to drop my rating weak accept.

---

> ### Author Response · Authors · 2024-08-14
>
> Dear reviewer PK9U, thank you for your comment. To further resolve your raised question, we provide the quantitative comparison with MVSplat [a] and MVSGaussians [b] as below:
>
> - PSNR (↑ The higher the better)
>
> |                 | Reconstruction (T) | Prediction (T + 5) | Prediction (T + 10) |
> | --------------- | ------------------ | -------------------| ------------------- |
> | **MVSplat**     | 21.84              | 20.14              | 18.78               |
> | **MVSGaussian** | 21.25              | 16.49              | 16.42               |
> | **SCube (Ours)**| 25.90              | 19.90              | 18.78               |
> | **SCube+ (Ours)**| **28.01**         | **22.32**          | **21.09**           |
>
> - SSIM  (↑ The higher the better)
>
> |                 | Reconstruction (T) | Prediction (T + 5) | Prediction (T + 10) |
> | --------------- | ------------------ | -------------------| ------------------- |
> | **MVSplat**     | 0.71               | 0.71               | 0.69                |
> | **MVSGaussian** | 0.80               | 0.70               | 0.60                |
> | **SCube (Ours)**| 0.77               | 0.72               | 0.70                |
> | **SCube+ (Ours)**| **0.81**          | **0.74**           | **0.72**            |
>
> - LPIPS (↓ The lower the better)
>
> |                 | Reconstruction (T) | Prediction (T + 5) | Prediction (T + 10) |
> | --------------- | ------------------ | -------------------| ------------------- |
> | **MVSplat**     | 0.46               | 0.48               | 0.52                |
> | **MVSGaussian**   | 0.51               | 0.60               | 0.59                |
> | **SCube (Ours)**| 0.45               | 0.47               | 0.49                |
> | **SCube+ (Ours)**| **0.25**          | **0.34**           | **0.38**            |
>
> For both methods, we use their respective official codebases and retrain the model using the same dataset split as our method with two 8-GPU nodes. We follow the recommended training hyperparameter and strategies to train the models. Please note that these methods are originally demonstrated only on indoor datasets with a smaller scale and without sky modeling.
>
> The above quantitative results echo our reasoning as posted before: While both methods show good reconstruction quality, they fail to model the true 3D geometry and the occluded parts of the scene and hence have unsatisfactory generalization capability to large-scale outdoor scenes.
>
> Geometry-wise, the geometry of MVSplat degenerates to multiple planes, and the geometry of MVSGaussian contains too many outliers on the rays of the corresponding pixels. To measure the geometric accuracy we compute the L2 Chamfer distance between predicted voxels and ground-truth voxel centers (note that the metric is in `meter`). The results are as follows:
>
> | Quantile               | 0.5 (median) | 0.6      | 0.7      | 0.8      | 0.9      |
> | ---------------------- | ------------ | -------- | -------- | -------- | -------- |
> | **MVSplat** |   43.61     |   45.54  |  47.47   |   49.67  |  53.77   |
> | **Ours**    | **0.10**     | **0.11** | **0.13** | **0.15** | **0.20** |
>
> *(Note that we do not include the results of MVSGaussian due to the amount of outliers reconstructed.)*
>
> Due to the restriction of NeurIPS, we have sent an anonymous link to the AC containing **qualitative** results of these baselines. Please ask the AC for the link if necessary. We will include the new comparisons to the final version of our paper and add corresponding citations.
>
> **We sincerely hope the supplementary experiments comparing to MVSGaussian and MVSplat have addressed your concerns, and we really appreciate it if you could reconsider our method and raise the score back.**
>
> ***References***:
>
> [a] Chen, Yuedong, et al. "Mvsplat: Efficient 3d gaussian splatting from sparse multi-view images." ECCV 2024.
>
> [b] Liu, Tianqi, et al. "Fast Generalizable Gaussian Splatting Reconstruction from Multi-View Stereo." arXiv preprint arXiv:2405.12218 (2024).

---

> > ### Author Response · Authors · 2024-08-14
> >
> > Dear reviewer,
> >
> > we sincerely hope the supplementary experiments comparing to MVSGaussian and MVSplat have addressed your concerns; and we really appreciate it if you could reconsider our method and raise the score back.

---

> > > ### Comment · Reviewer_PK9U · 2024-08-14
> > >
> > > Thank you very much for the experiment! I really appreciate it! I've raised my score back.

---

### Official Review · Reviewer_6vwn · 2024-07-13

**Soundness:** 3
**Presentation:** 3
**Contribution:** 3
**Rating:** 6
**Confidence:** 4

**Summary:**

The paper introduces a pipeline for street scene reconstruction given a sparse set of images as input. The reconstruction process follows a feed-forward manner. The method builds upon the existing XCube work. First, it generates sparse voxels of the scene to represent the geometries, then each voxel feature is decoded into Gaussian primitives for appearance rendering. Experimental results, both quantitative and qualitative, show that SCube can reconstruct the scene with high quality and provides benefits for downstream applications such as autonomous driving simulation.

**Strengths:**

(1) The results are impressive. Given the sparse images with little overlap, the reconstructed Gaussians are of high quality and show good generalization ability in novel views.

(2) Overall, the method sounds reasonable. Technical details are provided, and I believe the results are reproducible.

**Weaknesses:**

(1)	I am curious if we really need a diffusion model for this task. Since there are several images as conditions, the uncertainty (or randomness) of the output should be very small. Why not just train a regressor for the sparse voxel reconstruction? Is there any specific motivation for using a diffusion model?

**Questions:**

(1)	Why is the GAN loss used independently for each scene in the inference stage, instead of using it in the training stage?

**Limitations:**

Currently, the results are presented in the sparse view settings of a small region of the street scene. How can we reconstruct a very large scene using a long sequence of images in a feed-forward manner? This is an interesting topic for future work.

---

> ### Author Rebuttal · Authors · 2024-08-07
>
> Thank you very much for the constructive feedback and your positive comments on the results and technical contribution. We appreciate your effort in this process. In what follows we will quote your questions and try to resolve them.
>
> > **Necessity of a Diffusion Model.** I am curious if we really need a diffusion model for this task. Since there are several images as conditions, the uncertainty (or randomness) of the output should be very small. Why not just train a regressor for the sparse voxel reconstruction?
>
> We note that the uncertainty of the scene geometry given our input images is still large, and the problem that the model tackles is indeed non-trivial and sometimes even ill-posed. To demonstrate this, we compute the percentage of occluded voxels (that are invisible from the input images) w.r.t. all the ground-truth voxels, and the number is around **80%**. In addition to the generative modeling power, compared to a simple regressor, multi-step denoising diffusion models are empirically verified to have more data-fitting power [a].
>
> As a comparison specific to our task, we trained a single-step model that is conditioned on the input images and directly regresses the desired sparse voxel grid. To measure the geometric accuracy, we use a metric called 'voxel Chamfer distance' that measures the L2 Chamfer distance between predicted voxels and ground-truth voxels, divided by the voxel size. The results on Waymo Open Dataset are as follows:
>
>
> | Quantile               | 0.5 (median) | 0.6      | 0.7      | 0.8      | 0.9      |
> | ---------------------- | ------------ | -------- | -------- | -------- | -------- |
> | Simple regressor model | 15.46        | 19.12    | 22.61    | 34.81    | 52.57    |
> | Our diffusion model    | **0.26**     | **0.28** | **0.32** | **0.37** | **0.51** |
>
> As demonstrated in the above table, the diffusion model significantly outperforms a simple regressor.
>
> > **Sharing postprocessing GANs across datasets.** Why is the GAN loss used independently for each scene in the inference stage, instead of using it in the training stage?
>
> Thank you for the constructive suggestion -- we hereby present a postprocessing module that is jointly trained on the full dataset, without the need of per-scene finetuning. Specifically, we replace the original GAN with a more modern Img2img-Turbo model [b] and show the results in `Fig. B` in the PDF file. Note that while the essence idea of applying neural postprocessing on the rendered images stays the same, the new module removes the need for per-scene training for high-quality rendering. We will adopt this module in our revised version, and thanks for your suggestion again!
>
> > **Extending to longer input sequences.** Currently, the results are presented in the sparse view settings of a small region of the street scene. How can we reconstruct a very large scene using a long sequence of images in a feed-forward manner? This is an interesting topic for future work.
>
> We agree that this is an interesting and meaningful direction of future work and we present some preliminary results in `Fig. A` in the rebuttal PDF. We feed multiple frames of input images independently into our model and stitch the inference results from multiple timesteps together, without any finetuning. Results show that the stitched 3D scenes are temporally consistent despite minor artifacts, and one can similarly apply a longer LiDAR simulation session over the reconstruction. We also note that extending to even longer sequences could be implemented by recursively conditioning on past (latent) reconstructions, and we are actively exploring its feasibility.
>
>
> ***Reference:***
>
> [a] Yang et al. Diffusion models: A comprehensive survey of methods and applications. ACM Computing Surveys 2023.
>
> [b] Parmar et al. One-step image translation with text-to-image models. arXiv:2403.12036.

---

> ### Comment · Reviewer_6vwn · 2024-08-13
> **Comment**
>
> Thank you for the answers! They addressed my questions.

---

> > ### Author Response · Authors · 2024-08-14
> >
> > Thank you for taking the time to review our rebuttal!

---

### Official Review · Reviewer_CDsX · 2024-07-15

**Soundness:** 3
**Presentation:** 3
**Contribution:** 2
**Rating:** 4
**Confidence:** 4

**Summary:**

The paper proposes a new method to reconstruct 3D outdoor scenes from a sparse set of posed images. The key idea is to utilize a new hybrid 3D representation, which assigns 3D Gaussians to each sparse voxel. Given input images, the paper first adapts XCube to condition on sparse view images. Along with the generated fine-level voxels, the paper then designs a feed-forward model to reconstruct the appearance. This proposed pipeline can also be used in text-to-scene generation. Experiments show that the proposed method outperforms existing approaches.

**Strengths:**

Originality:

The paper is trying to solve a challenging problem: efficient large-scale scene reconstruction from sparse posed images. The motivation is to utilize the data prior to complement the sparse images. The proposed representation is a straightforward combination of XCube with 3D Gaussians (simply assigning 3D Gaussians into each voxel), which has also been widely applied (e.g., Scaffold-GS). While combining these two representations is acceptable, the concern is that the overall pipeline design seems more like a simple combination of two representations: first generating XCube, then generating 3D Gaussians with geometry & images. However, the problem to be solved is important, and this paper might inspire more researchers to work on this problem.

Quality:

The submission seems technically sound.

Clarity:

The paper is mostly clear and easy to read.

Significance:

Though there are some concerns with the methods and results, the paper is aiming at solving a very good problem.

**Weaknesses:**

- One concern is that the novelties of this paper are not very clear. As stated above (Originality), the paper seems to simply use two stages to combine the two representations in a simple way: one for XCube, and one for 3D Gaussians. It would be better if the author can clearly state the contributions.

- As a two-stage method, it would be better to discuss more about: (1) First stage: how to evaluate whether the reconstructed voxels align with the input images? (2) Second stage: how robust is the model when the input images and the voxels are not aligned?

- L187: Considering the temporal inputs, can the methods give consistent results temporally? Adding results might be better.

- L244-245: In Table 1, SCube seems to have very similar results to PixelSplat for future prediction. Why? An obviously better reconstruction result alone seems not able to show the effectiveness of the proposed method. The appearance reconstruction uses generated geometry (fine-level voxels) as input, and adds some tricks such as sky modeling. It is unknown how they affect the results.

**Questions:**

See Weaknesses

---

> ### Author Rebuttal · Authors · 2024-08-07
>
> Thank you for your constructive feedback, which helps us improve SCube. We are happy that you agree that the problem we solve is challenging and important, and the techniques are sound and well-written. In the following, we will quote your original comments and try to resolve them.
>
> > **Concerns about novelty.** The paper seems to simply use two stages to combine the two representations in a simple way: one for XCube, and one for 3D Gaussians. It would be better if the author can clearly state the contributions.
>
> Our main contribution is the novel framework that enables 3D priors for large-scale fast scene reconstruction that no prior work achieves. We humbly believe that our work adds new insights and knowledge into the literature to achieve ‘promising results’, as echoed by reviewer NKmV.
> Although at a high level, our method could be decomposed into two parts, we have made non-trivial technical contributions to synergize and tame them for the sparse-image-based 3D reconstruction task. This includes the image-based conditioning techniques to handle occlusions and 2D-3D alignments in the sparse diffusion architecture ($\S$ 3.1), the hybrid scene representation that facilitates both learning and fast rendering ($\S$ 3.2), along with effective training procedures ($\S$ 4.1). We remark that our framework significantly outperforms baseline approaches on the same tasks, enabling reconstructing scenes with $102.4\text{m}$ in scale with nice geometry under seconds. We also demonstrate a couple of useful applications available only with our proposed method ($\S$ 3.3).
> Hence, a re-evaluation of our novelty would be greatly appreciated.
>
> > **Alignments of voxels w.r.t. input images.** As a two-stage method, it would be better to discuss more about: (1) First stage: how to evaluate whether the reconstructed voxels align with the input images? (2) Second stage: how robust is the model when the input images and the voxels are not aligned?
>
> To quantitatively evaluate the pixel-voxel alignments (question (1)), we compute an additional metric called 'voxel Chamfer distance' that measures the L2 Chamfer distance between predicted voxels and ground-truth voxels (that are pixel-aligned), divided by the voxel size. This metric reflects the geometric accuracy of our prediction by measuring on average how many voxels is the prediction apart from the ground truth. The results on Waymo Open Dataset are as follows:
>
>
> | Quantile               | 0.5 (median) | 0.6  | 0.7  | 0.8  | 0.9  |
> | ---------------------- | ------------ | ---- | ---- | ---- | ---- |
> | Voxel Chamfer distance | **0.26**     | 0.28 | 0.32 | 0.37 | 0.51 |
>
> The table indicates that on 90% of the test samples, the predicted voxel grid is only half of a voxel off from the ground truth. We note that during our data curation process, there could be errors in the ground-truth voxels (due to, e.g., COLMAP failures), accounting for the outliers in the above metric.
> To answer question (2), we visualize the sample with the worst voxel Chamfer distance in `Fig. D` of the rebuttal PDF: we show that the predicted results are decent even though the ground truth is corrupted due to the lack of motion in the ego car. This also indicates that the voxels and the images are rarely misaligned, demonstrating the robustness of our method.
>
> > **Temporal inputs.** Considering the temporal inputs, can the methods give consistent results temporally? Adding results might be better.
>
> Thank you for your suggestion. We apply our full pipeline to a set of temporal inputs comprising of consecutive frames ($T$, $T+5$, $T+10$, $T+15$, $T+20$), and show the reconstruction and LiDAR simulation results in `Fig. A` in the rebuttal PDF. Our method is able to produce consistent 3D along the driving trajectory though the frames are fed independently.
>
> > **Future prediction results vs. PixelSplat.** In Table 1, SCube seems to have very similar results to PixelSplat for future prediction. Why? The appearance reconstruction uses generated geometry (fine-level voxels) as input and adds some tricks such as sky modeling. It is unknown how they affect the results.
>
> We kindly refer to Fig. 4 (2nd row) in the paper and the supplement video (at 00:43) for a 3D visualization of the reconstruction results for PixelSplat. The predicted geometry of PixelSplat is degenerated to planar patches, and the future rendering has a lot of black hole artifacts. In this particular scenario, PSNR is less sensitive to these artifacts and is admittedly not an ideal indicator of the quality, as echoed in [a, b], while LPIPS can capture such artifacts and faithfully reflect the results (Ours = 0.47 vs PixelSplat = 0.60 at $T+5$, where the lower the better).
> The appearance reconstruction module does require fine-level voxels as input otherwise it cannot determine the correct positions to generate the Gaussians. As our geometry of interest is limited to a square region around the ego vehicle, the VoxSplat representation cannot cover faraway regions such as the sky. Without sky modeling we cannot render the sky regions, and we show qualitative results in the supplementary video at, e.g., 00:09.
>
> ***Reference:***
>
> [a] Rockwell et al. Pixelsynth: Generating a 3d-consistent experience from a single image. ICCV 2021.
>
> [b] Ren et al. Look outside the room: Synthesizing a consistent long-term 3d scene video from a single image. CVPR 2022.

---

> > ### Comment · Reviewer_CDsX · 2024-08-13
> >
> > Thanks for the rebuttal. Most of my concerns are addressed but I am still concerned about the novelty.  I don't have follow-up questions within this rebuttal, and will have more discussion with other reviewers for the final decision.

---

> ### Author Response · Authors · 2024-08-14
>
> Thank you for taking the time to review our rebuttal. We are pleased that most of your concerns have been addressed.
>
> While our approach incorporates various state-of-the-art techniques such as Gaussian splatting and diffusion models, we recognize that many well-justified and published works also leverage one or more of these building stones to formulate their reconstruction models. Our primary contribution lies in the novel way of synergizing these foundational techniques to create a fully integrated pipeline that achieves state-of-the-art results for large-scale outdoor scene reconstruction encompassing true 3D geometric priors, while also introducing significant technical contributions necessary to make the method effective (as noted in the rebuttal above).
>
> Importantly, our method introduces the first possible way to tackle the challenging task of reconstructing large-scale 3D scenes from low-overlapping outward-facing input images. We are able to reach a much better and more robust performance on this task than the baselines (e.g. PixelSplat, DUSt3R, MVSGaussian, MVSplat, etc.) where a faithful reconstruction of the underlying 3D geometry is missing.
>
> We would greatly appreciate it if SCube's contribution could be reconsidered during the discussion phase.

---

### Author Rebuttal · Authors · 2024-08-07

We appreciate the insightful comments provided by the reviewers who all agree that SCube is technically sound and easy to understand, the results are impressive, and the problem solved is challenging. We post responses to reviewers individually in the corresponding section, while referenced figures are jointly included in the PDF file.

---

### Decision · Program_Chairs · 2024-09-25

**Decision:**

Accept (poster)

**Comment:**

This paper was reviewed by four experts in the field and received mixed reviews (1) Borderline Reject, (1) Weak Accept, (1) Accept, (1) Borderline Accept. Reviewers appreciate the idea of combining xcube diffusion model and gaussian splatting for sparse view-based large scale scene reconstruction. While one reviewer is concerned about the novelty, other reviewers find the idea is effective especially in reconstructing large scale scene from sparse views. The paper is well-written and justified with good results.  Reviewers have raised concerns regarding the details of  sky panorama model, evaluation of the results for both stages, and whether it is necessary to use diffusion for the task. Authors have provided responses to address reviewers’ concerns.  AC agrees with reviewers on the contribution of the idea which is effective and efficient in the reconstruction especially in handling the sparse view based reconstruction. AC thus recommends Accept.